# Differentiable Optimization Layers for Guaranteed Fairness in Deep Learning

**David Troxell** [1] **Noah Roemer** [1] **Guido Montúfar** [1 2 3]

## Abstract

Differentiable optimization layers are traditionally integrated in predict-then-optimize frameworks where a neural model estimates parameters that subsequently serve as fixed inputs to downstream decision-making optimization problems. In this work, we introduce the concept of a "fairness layer": a differentiable optimization layer appended to a model's output layer that guarantees a chosen notion of output parity is satisfied when integrated into a neural network. Additionally, we introduce an online primal-dual inference algorithm that provides provable aggregate fairness guarantees for streaming predictions with arbitrarily small batch sizes, where traditional per-batch constraints become overly restrictive. Numerical experiments demonstrate the effectiveness of the fairness layer and associated algorithm, and theoretical analysis characterizes the layer's differentiability and stability properties during model training and backpropagation. Our code for these experiments is publicly available on GitHub: `https://github.com/dtroxell19/FairDL-ICML-2026.git` and our public Python package documentation can be found online: `https://dtroxell19.github.io/fairness_training/`.

## 1. Introduction

### 1.1. Motivation

Machine learning systems are increasingly being deployed in high-stakes scenarios including loan applications, hiring, court decisions, clinical diagnoses, and autonomous vehicles (Galindo & Tamayo, 2000; Van den Broek et al., 2021; Berk, 2017; Bakator & Radosav, 2018; Gupta et al., 2021). Consequently, the design and evaluation of "fair" models has become a focal point of research globally. Additionally, regulatory efforts have increased, such as the European Union's 2024 AI Act (European Parliament and Council of the European Union, 2024) and US federal and municipal bias audit laws (Official Website of the City of New York, 2023; United States EEOC Press Release, 2021). To meet these demands, numerous fairness definitions have been proposed and studied, with this work focusing on "group-fairness" constraints that enforce equality of statistical measures across protected groups.

### 1.2. Background

**Fairness in Deep Learning.** Methods for mathematically incorporating group-fairness or individual-fairness constraints into deep learning pipelines can be categorized into three groups: pre-processing, in-processing, and post-processing methods (Du et al., 2021).

Pre-processing methods modify or augment training data prior to model learning. For example, diffusion models have been employed to augment datasets with realistic medical image scans from underrepresented groups, thus mitigating model bias (Ktena et al., 2024). In-processing methods attempt to ensure fairness during model training via Lagrangian penalty terms in loss functions or adversarial methods to prevent protected attribute encoding in latent representations (for example Padala & Gujar, 2021; Delobelle et al., 2021). Post-processing methods directly modify predictions made by a network after training. Examples include projecting the original predictions onto a constraint set or post-hoc algorithms reducing accuracy discrepancies across subgroups (Wei et al., 2020; Kim et al., 2019).

Each approach has limitations: pre-processing can amplify bias (Wang et al., 2018), in-processing fails to guarantee constraint satisfaction due to surrogates or soft penalties, and post-processing operates on fairness-unaware models that may not generalize after constraint enforcement.

**Differentiable Optimization Layers.** A differentiable optimization layer (often referred to as a "declarative node") is a neural network component whose forward pass is implicitly defined as the solution to an optimization problem rather than explicitly defined by a prescribed activation function

[1]Department of Statistics & Data Science, University of California, Los Angeles, USA [2]Department of Mathematics, University of California, Los Angeles, USA [3]Max Planck Institute for Mathematics in the Sciences, Leipzig, Germany. Correspondence to: David Troxell <davidtroxell@ucla.edu>.

*Proceedings of the $43^{rd}$ International Conference on Machine Learning*, Seoul, South Korea. PMLR 306, 2026. 

(Gould et al., 2021). Gradients can be computed via the implicit function theorem and KKT conditions (Karush, 1939; Kuhn & Tucker, 1951), and efficient implementations exist via disciplined convex programming (Grant et al., 2006; Agrawal et al., 2019).

**End-to-End Modeling.** Declarative nodes enable optimization-based reasoning in "predict-then-optimize" frameworks, where neural networks estimate parameters for downstream optimization problems. End-to-end differentiability allows learning decision variables directly from data, with applications in optimal control, portfolio optimization, and energy scheduling (Agrawal et al., 2020; Uysal et al., 2024; Sun et al., 2023).

### 1.3. Goals for Advancement in Fair Deep Learning

We aim to introduce a new component for deep learning architectures that satisfies the following goals: (G1) Verified Fairness: strict compliance to a pre-specified level of fairness; (G2) End-to-End Learning: maintains differentiability through constraints and does not rely on post-hoc adjustments after training; (G3) High Flexibility: compatible with any architecture.

Existing methods satisfy only subsets of these goals (See Appendix G for details): pre-processing (Rajabi & Garibay, 2021; Kamiran & Calders, 2012) and in-processing (Adel et al., 2019; Delobelle et al., 2021; Padala & Gujar, 2021) methods achieve (G2) and (G3) but not (G1). Post-processing (Wei et al., 2020; Kim et al., 2019) can achieve (G1) and (G3) but not (G2). To the best of our knowledge, no prior work simultaneously achieves all three.

### 1.4. Contributions

1. We introduce the "fairness layer": a flexible architectural component guaranteeing fairness while maintaining end-to-end trainability; to the author's knowledge, the first method satisfying (G1), (G2), and (G3) for affine constraint sets.

2. We propose an online primal-dual inference algorithm overcoming the fundamental challenge of enforcing fairness with arbitrarily small batch sizes, providing deterministic aggregate guarantees where per-batch constraints are prohibitively restrictive.

3. We conduct experiments demonstrating improvements over Lagrangian penalties and post-hoc projections. We created the `fairness_training` Python package for practitioners interested in the fairness layer.

4. We provide mathematical analysis of stability and differentiability properties of the fairness layer that motivate its practical utility.

## 2. The Fairness Layer

Rather than deploying differentiable optimization layers in a typical predict-then-optimize framework, we propose a differentiable "fairness layer" appended to the network's output layer. This fairness layer projects raw outputs onto a fairness constraint set.

Let $\mathcal{D}_{\text{train}} = \{(x_i, y_i)\}_{i=1}^n$ denote a training dataset with labels $y_i \in \mathbb{R}\ \forall i$ and inputs $x_i \in \mathbb{R}^d\ \forall i$. We use $X = [x_1, \ldots, x_n]^\top \in \mathbb{R}^{n\times d}$ as shorthand notation for the training dataset of inputs and $x_{ij}$ to denote the $j$-th attribute of observation $i$. Additionally, let $f_\theta(x) : \mathbb{R}^d \mapsto \mathbb{R}$ denote a neural network such that $f_\theta(x) = (f_L \circ f_{L-1} \circ \cdots \circ f_1)(x)$ where the intermediate layers are defined as $f_l(h) = a(W^{(l)}h + b^{(l)}),\ l = 1, \ldots, L-1$ and $a$ is some activation function applied elementwise. For a given input batch $X^{(b)}$ with $n_b$ observations, denote the $j$-th feature vector as $X^{(b)}_{:j}$ and the batched output of the network as $z \triangleq f_\theta(X^{(b)}) = [f_\theta(x_1), \ldots, f_\theta(x_{n_b})]^\top \in \mathbb{R}^{n_b}$. For given constraint functions $h_{\text{ineq}} \colon \mathbb{R}^{n_b} \to \mathbb{R}^q$ and $h_{\text{eq}} \colon \mathbb{R}^{n_b} \to \mathbb{R}^v$ and discrepancy function $\tilde{d} \colon \mathbb{R}^{n_b} \times \mathbb{R}^{n_b} \to \mathbb{R}$, we define the general fairness layer as a function mapping $z$ to

$$g(z) = \operatorname*{arg\,min}_{\tilde{y}\in\mathbb{R}^b} \{\tilde{d}(\tilde{y}, z) \colon h_{\text{ineq}}(\tilde{y}) \le 0, h_{\text{eq}}(\tilde{y}) = 0\}. \quad (1)$$

In this work, we restrict to the case where $d(\tilde{y}, z)$ is convex in $\tilde{y}$ and the constraint set is defined by linear equations and linear inequalities $h_{\text{ineq}}(\tilde{y}) = A\tilde{y} - m_1$ and $h_{\text{eq}}(\tilde{y}) = B\tilde{y} - m_2$ for some $A \in \mathbb{R}^{q\times n_b}, m_1 \in \mathbb{R}^q$ and $B \in \mathbb{R}^{v\times n_b}, m_2 \in \mathbb{R}^v$. In principle, however, the definitions extend naturally to more general cases. Regardless, the final batched output of the network is denoted as:

$$\hat{y} = g(f_\theta(X^{(b)})) \in \mathbb{R}^{n_b}.$$

Thus the fairness layer maps a batch of raw predictions to the closest possible batch of predictions satisfying the desired constraints. Note that the choice of dissimilarity function $\tilde{d}(\cdot,\cdot)$ is left to the modeler. Additionally, the constraint set in (1) must be defined by the modeler to reflect the desired fairness criteria in a given environment. Please see Appendix A for a discussion on differentiation through the fairness layer.

In the context of model fairness, we designate some of the features $j \in [d]$ of an input dataset as "protected attributes", which we will assume to be binary-valued, with $1$ denoting membership in the group of interest and $0$ denoting non-membership for a given observation. We use $\mathcal{S} \subseteq [d]$ to denote the set of features corresponding to protected attributes in the dataset. Importantly, each protected attribute is considered independently; a single observation can belong to multiple protected groups (one per feature) rather than a single group defined jointly by all protected attributes.

Many commonly used notions of fairness for both regression and classification can be formulated as affine constraints of the form given in (1). Specifically, we consider fairness criteria that require the equality of expectations of relevant quantities, conditioned on specified regions of the input or output spaces:

1. Expected Conditional (or Demographic) Parity (Ritov et al., 2017), or Mean (Conditional) Parity (Wei et al., 2023) or Mean (Conditional) Difference (Calders et al., 2013):

$$-\epsilon \leq \mathbb{E}[\tilde{y}|X_{:j^*} = 0, X_{:j} \in \mathcal{R}_j] - \mathbb{E}[\tilde{y}|X_{:j^*} = 1, X_{:j} \in \mathcal{R}_j] \leq \epsilon \quad \forall\, j^* \in \mathcal{S} \text{ and } j \in [d] \setminus \mathcal{S}, \tag{2}$$

   where $\mathcal{R}_j$ specifies the considered range of the $j$-th feature of the observation, and $\epsilon$ is a small tolerance threshold. This notion of fairness ensures that, given some input feature values are in similar ranges, the average rating or prediction for two groups is similar.

2. Expected Equalized Residuals (Grari et al., 2020):

$$-\epsilon \leq \mathbb{E}[\tilde{y} - y|X_{:j} = 0] - \mathbb{E}[\tilde{y} - y|X_{:j} = 1] \leq \epsilon \quad \forall j \in \mathcal{S}. \tag{3}$$

   This notion of fairness ensures that the model overestimates or underestimates each group's outcome in a similar fashion on average.

3. Expected Equalized Odds (Hardt et al., 2016):

$$-\varepsilon \leq \mathbb{E}[\tilde{y}|y \in \mathcal{I}_i, X_{:j} = 0] - \mathbb{E}[\tilde{y}|y \in \mathcal{I}_i, X_{:j} = 1] \leq \varepsilon \quad \forall j \in \mathcal{S} \text{ and } \forall\, i. \tag{4}$$

   Here $\mathcal{I}_i$ represents a region of the output space. In the case of binary classification, for instance, $\mathcal{I}_0 = \{0\}$ and $\mathcal{I}_1 = \{1\}$. This notion of fairness ensures that individuals with similar target values are treated similarly across groups.

As discussed in Section 3, the expectations can be approximated via sample means over minibatches. Note that in classification settings, many classical notions of fairness such as demographic parity and equalized odds are computed using hard model assignments, resulting in nonconvex formulations (see Appendix B for details). In this work, we replace the nonconvex entities with expectations, yielding convex constraints. While these expectation-based constraints can be used as tractable proxies for classical notions of fairness in classification settings, we note that expectation-based constraints often align exactly with established desideratum in regression, scoring, and continuous prediction settings, and can also be applied to pre-thresholded classification logits. Additionally, although this work emphasizes constraints for fairness, the general fairness layer formulation (1) also naturally accommodates other constraints not related to fairness (see Appendix C for examples).

## 3. Algorithms and Minibatches

Ultimately, the goal is to obtain a model whose inference predictions satisfy population-level constraints. In practice, however, models are deployed with finite batch sizes. Because the output of the fairness layer $g(\cdot)$ is batch-dependent, the choices of batch sizes for training $b_{\text{train}}$ and inference $b_{\text{infer}}$ have large implications in the fairness setting. Let $\mathcal{F}(\hat{y}_{b,i}; X^{(b,i)})$ denote an affine statistic computed over the predictions and features for group $i \in \{0, 1\}$ within minibatch $b$ (e.g., the sample mean prediction $\frac{1}{n_{b,i}} \sum_{j:x_{jk}=i} \hat{y}_j$ for protected attribute $k \in \mathcal{S}$). We use $\mathcal{F}_{b,i}$ as shorthand. The fairness constraints enumerated in (2)–(4) can then be expressed as bounds on the gap between group statistics: $|\mathcal{F}_0 - \mathcal{F}_1| \leq \epsilon$. Define the weighted fairness violation as

$$w_b(\hat{y}_b) := n_b \cdot \big(|\mathcal{F}_0(\hat{y}_b; X^{(b)}) - \mathcal{F}_1(\hat{y}_b; X^{(b)})| - \epsilon\big).$$

The following lemma details the conditions when enforcing fairness per mini-batch results in fairness constraints being satisfied across all samples in aggregate:

**Lemma 3.1** (Aggregate Fairness Under Varying Group Proportions)**.** *Given $B$ batches of input data, let $\mathcal{F}_{b,i} = \mathcal{F}(\hat{y}_{b,i}; X^{(b,i)})$ denote an affine fairness statistic for group $i \in \{0, 1\}$ in batch $b \in [B]$. Let $n_{b,i}$ denote the number of samples from group $i$ in batch $b$.*

*Suppose each batch $b$ satisfies the per-batch fairness constraint:*

$$|\mathcal{F}_{b,0} - \mathcal{F}_{b,1}| \leq \epsilon.$$

*Let $p_b = \frac{n_{b,0}}{n_{b,0}+n_{b,1}}$ denote the proportion of group 0 in batch $b$, and let $\bar{p} = \frac{\sum_b n_{b,0}}{\sum_b (n_{b,0}+n_{b,1})}$ denote the overall proportion of group 0 across all batches.*

*Define the aggregate fairness statistics:*

$$\mathcal{F}_0 = \frac{\sum_b n_{b,0}\mathcal{F}_{b,0}}{\sum_b n_{b,0}}, \quad \mathcal{F}_1 = \frac{\sum_b n_{b,1}\mathcal{F}_{b,1}}{\sum_b n_{b,1}}.$$

*Then the aggregate fairness satisfies:*

$$|\mathcal{F}_0 - \mathcal{F}_1| \leq \epsilon + \Delta_p \cdot R \left(\frac{1}{\bar{p}} + \frac{1}{1 - \bar{p}}\right),$$

*where:*

- $\Delta_p = \max_b |p_b - \bar{p}|$ *measures the maximum deviation of batch group proportions from the overall proportion.*

- $R = \max_{b,i} |\mathcal{F}_{b,i}|$ *is the maximum absolute fairness statistic value across all batches and groups.*

The proof is given in Appendix H.1. In practice, stratified sampling can enforce consistent group proportions across minibatches, in which case the lemma guarantees satisfaction of constraints with target tolerance $\epsilon$. However, in real-time or streaming prediction settings, maintaining consistent group membership across batches may be infeasible. While $\Delta_p$ vanishes with large batch sizes, small inference batches introduce challenges: even when constant group proportions can be enforced, minibatch-level fairness constraints may overly restrict model expressivity. To address these limitations, we propose an online primal-dual optimization algorithm that permits individual minibatches to violate fairness constraints while guaranteeing aggregate fairness over time.

At each inference step in the small-batch inference regime, predictions are obtained by solving a differentiable constrained optimization problem whose objective includes a dynamically updated fairness penalty. Let $t = 1, 2, \dots, T$ index time such that, at time $t$, batch $b_t$ arrives for inference. The model produces predictions $f_\theta(X^{(b_t)}) = \hat{y}_{b_t} \in \mathbb{R}^{|b_t|}$ where $|b_t|$ is used as shorthand notion to denote the size of batch $b_t$. We set $b_{\text{infer}} = \max_t |b_t|$.

Given a target fairness level $\epsilon > 0$, define the batch-level fairness gap and violation, respectively, as:

$$v_t(\hat{y}_{b_t}) := |\mathcal{F}_{b_t,0} - \mathcal{F}_{b_t,1}|,$$
$$\tilde{v}_t(\hat{y}_{b_t}) := v_t(\hat{y}_{b_t}) - \epsilon.$$

We therefore define the weighted violation as follows:

$$w_t(\hat{y}_{b_t}) \triangleq |b_t| \cdot \tilde{v}_t(\hat{y}_{b_t}).$$

Our objective is to satisfy the global fairness constraint

$$\sum_{t=1}^{T} w_t(\hat{y}_{b_t}) \leq 0 \;\Leftrightarrow\; \frac{1}{\sum_{t=1}^{T} |b_t|} \sum_{t=1}^{T} |b_t| v(t) \leq \epsilon. \tag{5}$$

Let $b_\tau$ denote a threshold batch size above which stratified sampling is feasible; that is, one can form mini-batches that have at least one observation per protected group, and group composition is constant across minibatches. In Section 5, $b_\tau = 2000$ and $b_\tau = 256$ are used, but this choice is left to the modeler. Algorithm 1 enforces fairness as a cumulative constraint across all inference batches. While individual batches—particularly when $b_{\text{infer}}$ is small—may violate fairness constraints, Theorem 3.2 guarantees that the sample-weighted average violation converges to at most $\epsilon$ as the number of inference batches grows. The following theorem formalizes this guarantee:

**Algorithm 1** Primal-Dual Constraint-Aware Inference for Fairness

**Input:** Trained model $f_\theta$, fairness layer $g$, stream of inference batches $\{X^{(b_t)}\}\ \forall t \in [T]$, fairness tolerance $\epsilon$, dual step size $\eta$, batch threshold $b_\tau$
**Initialize:** Dual variable $\lambda \leftarrow 0$, dual update count $t \leftarrow 0$
**for** each inference batch $X^{(b_t)}$ of size $b_{\text{infer}}$ **do**
  Compute raw predictions: $\hat{y}_{\text{raw}} \leftarrow f_\theta(X^{(b_t)})$
  **if** $b_{\text{infer}} < b_\tau$ **then**
    Compute adaptive step size: $\eta_t \leftarrow \eta/\sqrt{t+1}$
    **Primal update (primal-dual projection):**

$$\hat{y}_{b_t} \leftarrow \arg\min_{\hat{y}} \left[\|\hat{y} - \hat{y}_{\text{raw}}\|^2 + \lambda \cdot b_{\text{infer}} \cdot v(\hat{y})\right] \tag{6}$$

    Compute fairness violation:

$$v_{b_t} \leftarrow b_{\text{infer}} \cdot \big(v(\hat{y}_{b_t}) - \epsilon\big)$$

    **Dual update:**

$$\lambda \leftarrow \max\{0, \lambda + \eta_t w_b\}$$

    $t \leftarrow t + 1$
  **else**
    **Hard constraint projection (per-batch fairness):**

$$\hat{y}_{b_t} \leftarrow \arg\min_{\hat{y}} \|\hat{y} - \hat{y}_{\text{raw}}\|^2 \quad \text{s.t.} \quad v(\hat{y}) \leq \epsilon \tag{7}$$

  **end if**
  **Output:** Return predictions $\hat{y}_{b_t}$ for batch $X^{(b_t)}$
**end for**

**Theorem 3.2** (Aggregate Fairness in Small-Batch Regime)**.** *Let $\ell_t(\hat{y}_{b_t}, y_{b_t})$ denote a differentiable loss function convex in $\hat{y}_{b_t}$ incurred at time $t$ for all $t$, and assume there exists a sequence $\{\tilde{y}_{b_t}\}_{t=1}^{\infty}$ such that $w_t(\tilde{y}_{b_t}) \leq 0$ for all $t$. Then, the sequence $\{\hat{y}_{b_t}\}$ produced by (6)–(7) satisfies*

$$\limsup_{T\to\infty} \frac{1}{\sum_{t=1}^{T} |b_t|} \sum_{t=1}^{T} |b_t| v(\hat{y}_{b_t}) \leq \epsilon.$$

The proof is detailed in Appendix H.2. In practice, this result implies that arbitrarily small batch sizes can be used during inference while controlling the batch-size-weighted time-average of fairness violations, provided sufficiently many batches are processed. The bound is asymptotic, so more inference batches yield a tighter approximation to the target fairness level $\epsilon$. When group proportions are stable across batches, this time-average coincides with the pooled aggregate fairness gap.

## 4. Properties of the Fairness Layer

While some structural properties of differentiable convex optimization layers have been established in other works (e.g., Gould et al., 2021), our aim in this section is to present results tailored to the context of fairness and the definition of the fairness layer described in Section 2. By articulating the correspondence between active constraint patterns, affine regions, and the resulting backward pass, we aim to provide an analysis that is both more transparent and more directly interpretable in context of the fairness layer.

**Theorem 4.1** (Differentiability of the fairness layer)**.** *Let $\mathcal{C} \subset \mathbb{R}^n$ be a nonempty, closed, convex set. Let $h : \mathbb{R}^n \to \mathbb{R}$ be $\mu$-strongly convex for some $\mu > 0$. Define*

$$g(z) := \arg\min_{\tilde{y} \in \mathcal{C}} \left\{ h(\tilde{y}) - \langle z, \tilde{y} \rangle \right\}, \quad z \in \mathbb{R}^n.$$

*Then for every $z \in \mathbb{R}^n$, $g$ is globally $1/\mu$-Lipschitz continuous and differentiable almost everywhere.*

Note that this objective function is a special case of the general discrepancy function (1). However, this form still encapsulates many common choices, such as the Euclidean discrepancy function $\tilde{d}(\tilde{y}, z) = \|\tilde{y} - z\|_2^2$.

Theorem 4.1 details that the inclusion of the fairness component in a neural network architecture will not disrupt backpropagation as the gradients are well-defined since the fairness component $g(z)$ is differentiable almost everywhere. However, despite being well defined, unstable or large gradients can cause gradient explosions or instability in training. The following corollary states that for certain choices of $h$, the inclusion of the fairness layer will certifiably not cause exploding gradients during model training.

**Corollary 4.2** (Gradient Stability for Fairness Layers)**.** *Let $g(z)$ be the fairness layer defined in Theorem 4.1:*

$$g(z) := \arg\min_{\tilde{y} \in \mathcal{C}} \left\{ h(\tilde{y}) - \langle z, \tilde{y} \rangle \right\},$$

*where we assume that $h$ is $\mu$-strongly convex. Then, $\|\mathbf{D}g(z)\|_2 \leq \frac{1}{\mu}$ almost everywhere, where $\mathbf{D}g(z)$ denotes the Jacobian matrix. Additionally, for any differentiable upstream map $f$:*

$$\|\nabla_X (g \circ f)(X)\|_2 \leq \frac{1}{\mu} \|\nabla_X f(X)\|_2.$$

These results suggest that the differentiable optimization layer is well-suited for model training in practical settings.

**Theorem 4.3** (Structure of the Fairness Layer)**.** *Let $g(z)$ be the fairness layer from Theorem 4.1 with $h(y) = \frac{1}{2}\|y\|_2^2$:*

$$g(z) = \arg\min_{y \in \mathcal{C}} \frac{1}{2} \|y - z\|_2^2, \tag{8}$$

*where $\mathcal{C} = \{y \in \mathbb{R}^n : Ay \leq b\}$ with $A \in \mathbb{R}^{m \times n}$ and $b \in \mathbb{R}^m$. For any $z \in \mathbb{R}^n$, define the index set of active constraints:*

$$\mathcal{A}(z) = \{i \in \{1, \ldots, m\} : (Ag(z))_i = b_i\},$$

*and let $A_{\mathcal{A}} \in \mathbb{R}^{|\mathcal{A}| \times n}$ denote a matrix constructed with rows of $A$ indexed by $\mathcal{A}(z)$. For any $z \in \mathbb{R}^n$, let $\lambda(z) \in \mathbb{R}^m$ denote the vector of KKT multipliers satisfying the optimality conditions:*

$$\begin{gathered} g(z) - z + A^\top \lambda(z) = 0, \quad \lambda_i(z) \geq 0, \\ \lambda_i(z)(Ag(z) - b)_i = 0. \end{gathered} \tag{9}$$

*Then:*

1. *Partition: The domain $\mathbb{R}^n$ can be partitioned into finitely many polyhedral regions $\{R_{\mathcal{I}}\}_{\mathcal{I} \subseteq \{1,\ldots,m\}}$,*

   $$R_{\mathcal{I}} = \{z \in \mathbb{R}^n : \mathcal{A}(z) = \mathcal{I} \text{ and } \lambda_i(z) > 0 \; \forall i \in \mathcal{I}\}$$

2. *Local structure: On each polyhedral region $R_{\mathcal{I}}$, the function $g$ is affine:*

   $$g(z) = P_{\mathcal{I}} z + c_{\mathcal{I}},$$

   *where $P_{\mathcal{I}} = I - A_{\mathcal{I}}^\top (A_{\mathcal{I}} A_{\mathcal{I}}^\top)^+ A_{\mathcal{I}}$ and $c_{\mathcal{I}} = A_{\mathcal{I}}^\top (A_{\mathcal{I}} A_{\mathcal{I}}^\top)^+ b_{\mathcal{I}}$ is a constant vector.*

3. *Measure zero boundaries: The boundary between any two regions $R_{\mathcal{I}}$ and $R_{\mathcal{J}}$ ($\mathcal{I} \neq \mathcal{J}$) has Lebesgue measure zero in $\mathbb{R}^n$.*

Property 3 from Theorem 4.3 verifies the efficacy of optimizers that stochastically sample batches: since the boundaries have measure 0, the probability of an iterate landing on the boundary is 0. Therefore, the optimizers do not get stuck as they will not land exactly on a boundary. Note that at a boundary between regions $R_{\mathcal{I}}$ and $R_{\mathcal{J}}$, the Jacobian $\nabla g(z)$ changes discontinuously from $P_{\mathcal{I}}$ to $P_{\mathcal{J}}$. An optimizer landing exactly on such boundaries would encounter an ill-defined gradient (the subdifferential would be multi-valued), potentially causing oscillations or failure to make progress.

**Theorem 4.4** (Spectral Structure and Gradient Supression of Fairness Layer Jacobian)**.** *Let $g(z)$ and $\mathcal{A}(z)$ denote the instantiation of the fairness layer and index set of active constraints as defined in Theorem 4.3. Then, by Theorem 4.1, $g$ is globally Lipschitz continuous with constant 1 and differentiable almost everywhere. At points where $g$ is differentiable, the Jacobian $\mathbf{D}g(z) \in \mathbb{R}^{n \times n}$ satisfies:*

1. ***Binary spectrum**: All eigenvalues of $\mathbf{D}g(z)$ belong to $\{0, 1\}$.*

2. ***Gradient suppression**: The component of any gradient normal to the active constraint surface is completely suppressed. In the context of backpropagation, for any gradient vector $v \in \mathbb{R}^n$:*

$$A_{\mathcal{A}}(\mathbf{D}g(z) \cdot v) = 0.$$

Theorem 4.4 provides an interpretation of the gradients induced by the fairness layer: any gradient in the "unfairness direction" (i.e., directions normal to the active constraint surface defined by $\text{span}\{a_i^T : i \in \mathcal{A}(z)\}$) will be set exactly to 0, while gradients in all other directions are unaffected. This ensures that backpropagation never encourages the model to move in directions that would violate active fairness constraints, while allowing learning to proceed freely in all feasible directions.

# 5. Numerical Experiments

We conduct numerical experiments across four distinct dataset settings and fairness constraint formulations to evaluate the effectiveness of the fairness layer and proposed inference algorithm. To highlight the versatility of the fairness layer, we demonstrate its use in two distinct paradigms: data modeling and predictive inference (Breiman, 2001).

A data modeling task assumes a parametric form for the joint distribution of $(x, y)$ and seeks to infer the parameter vector governing this process. Rather than pure predictive accuracy, the emphasis is on understanding structural relationships and parameter interpretability. In contrast, the primary objective in a predictive inference task is to construct a mapping $f\colon \mathcal{X} \to \mathbb{R}$ that minimizes expected prediction error, and the emphasis lies in generalization.

## 5.1. Models

While the specific fairness constraints and instantiations of the fairness layer $g(\cdot)$ vary for each numerical experiment setting, similar methods are employed for comparison across all datasets:

The first method utilizes the differentiable optimization fairness layer component $g(z)$ described in Section 2. The model is trained with Stochastic Gradient Descent (SGD), and for predictive inference tasks, Algorithm 1 is employed. In the subsequent sections, we call this method the "F-Layer" method. The F-Layer method's final output is defined as $\hat{y} = g(f_\theta(X))$.

The "Projection" baseline trains a neural network using standard SGD without fairness considerations, then projects predictions onto the constraint set post-hoc during inference after training is complete. The "Penalty" and "Strict Penalty" baselines minimize an augmented loss with constraint violation penalties:

$$\mathcal{L}_{\text{penalty}} = \frac{1}{N}\sum_{i=1}^{N} \ell(\hat{y}_i, y_i) + \lambda \tilde{\ell}_{\mathcal{C}},$$

where $\ell(\hat{y}_i, y_i)$ is standard MSE or cross-entropy loss, $\tilde{\ell}_{\mathcal{C}}$ measures constraint violation (typically quadratic or absolute value penalty), and $\lambda > 0$ is a weighting hyperparameter. The "Penalty" method uses cross-validation to select the smallest $\lambda$ satisfying fairness constraints, while "Strict Penalty" uses $\lambda \gg 0$ to likely ensure constraint satisfaction during inference.

## 5.2. Loan Default Prediction

This first experimental setting concerns loan default predictions for small businesses. Note for any given random train, validation, and test split, the dataset sizes are relatively small. Therefore, we repeat the entire experiment (including $\lambda$ selection for penalty models) 25 times, each with a new, random dataset split to reduce the impact of any given random data split on the reported results. See Appendix D for dataset details.

### 5.2.1. Constraints

In the loan default classification experiments, the fairness layer is defined as follows:

$$\begin{aligned} g(z) = &\arg\min_p \|p - z\|_2^2 \\ \text{s.t.} &\left| \frac{1}{n - \sum_i x_{ij}} \sum_{\{i\,:\,x_{ij}=0\}} p_i \right. \\ &\qquad \left. - \frac{1}{\sum_i x_{ij}} \sum_{\{i\,:\,x_{ij}=1\}} p_i \right| \le 0.01 \quad \forall j \in \mathcal{S}. \end{aligned} \tag{10}$$

Here $z$ are the raw predicted logits from the model. In the experiments, $|\mathcal{S}| = 2$. The first protected attribute records whether the business is in a rural or non-rural area, and the second attribute denotes whether the business is newly created or not. These constraints ensure that the model, on average, does not discriminate against new business or businesses in rural areas. The F-Layer and Projection methods employ the projection defined in (10), while Strict Penalty minimizes an unconstrained, regularized version of the loss function (see Appendix E).

### 5.2.2. Results

First, we observe in Figures 1a, 1b, and 1c that the F-Layer method yields modest but consistent improvements in AUC over the Projection method; the mean percent improvement offered by the F-Layer method is $1.46\%$. In the Precision-Recall curve, however, we see regions where the F-Layer

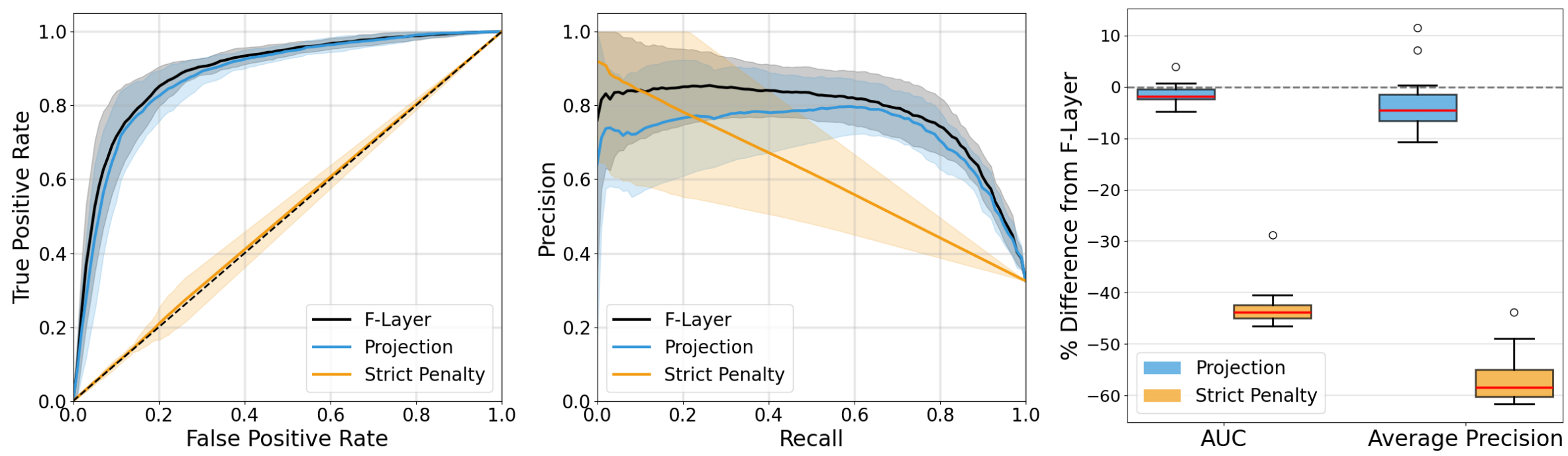

*(a)* *(b)* *(c)*

*Figure 1.* (a) ROC curves on the test set, averaged over 25 experiment repetitions with different, random train and test splits (with shaded regions representing $\pm 1$ standard deviation); (b) Precision-Recall curves on the test set, averaged over 25 experiment repeats (with shaded regions representing $\pm 1$ standard deviation); (c) Distribution of AUC and average precision on test set for all 25 experiment repetitions.

method largely outperforms the Projection method. We also find that the Strict Penalty method exhibits degenerate behavior; the model collapses to predicting a single class for every observation. This results in an average reduction of approximately 45% in AUC and 55% in average precision. Importantly, all 3 methods result in all constraints being satisfied for every random data split. However, the F-Layer method consistently outperforms other methods when balancing precision and recall.

### 5.3. Employee Performance Data

Next, we explore the use-case of a company utilizing a machine learning model to analyze employee performance data. The model is tasked to predict the hourly wage of each employee, and any large, negative residuals $r_i = y_i - \hat{y}_i$ are flagged to management, indicating the employee is potentially underpaid and deserves a wage increase. See Appendix D for dataset details.

#### 5.3.1. Constraints

We define the fairness layer as follows:

$$g(z) = \arg\min_{p} \|p - z\|_2^2$$
$$\text{s.t.} \begin{cases} \left|\frac{1}{n - \sum_i x_{ij}} \sum_{\{i\,:\,x_{ij}=0\}} p_i - y_i\right| \leq 10^{-4} \; \forall j \in \mathcal{S} \\ \left|\frac{1}{\sum_i x_{ij}} \sum_{\{i\,:\,x_{ij}=1\}} p_i - y_i\right| \leq 10^{-4} \quad \forall j \in \mathcal{S} \\ 0 \leq p_i \leq 1 \quad \forall i. \end{cases} \tag{11}$$

Here $y_i$ is the ground truth target for sample $i$ and $z = f_\theta(X^{(b)}) \in \mathbb{R}^{n_b}$. See Appendix E for the corresponding Penalty and Strict Penalty regularized loss formulations. We use $|\mathcal{S}| = 5$ protected attributes: the employee's managerial status (manager versus not a manager), age (at least 50 years old versus younger), gender (male versus female), degree category (technical degree obtained versus other or no degree), and marital status (married versus not married).

*Table 1.* Comparison of different methods for the data modeling task of employee raise recommendation.

| Model | Constraints Satisfied | MSE | Individuals Lowered |
|---|---|---|---|
| F-Layer | 10/10 | 0.0834 | N/A |
| Projection | 10/10 | 0.0838 | 319 |
| Penalty ($\lambda = 100$) | 10/10 | 0.0933 | N/A |
| Penalty ($\lambda = 10$) | 9/10 | 0.0938 | N/A |
| Penalty ($\lambda = 1$) | 1/10 | 0.0924 | N/A |
| Penalty ($\lambda = .01$) | 1/10 | 0.0827 | N/A |

#### 5.3.2. Results

Since this is a data modeling task, we evaluate models on three criteria: satisfying fairness constraints (11), achieving reasonable error as wage prediction residuals are meaningless otherwise, and avoiding post-hoc adjustments to avoid substantially altering original predictions to account for fairness, thereby decreasing transparency and trust in the modeling process.

Table 1 shows Penalty models with $\lambda < 100$ fail to satisfy constraints. Among constraint-satisfying methods, F-Layer and Projection achieve similar MSE (0.0834 vs 0.0838), while Penalty ($\lambda = 100$) has 11.2% higher error. However, Projection modifies predictions post-hoc, lowering wages for 319/1200 individuals (up to 5%), which reduces transparency. The F-Layer method satisfies all constraints, achieves lowest error, and incorporates fairness end-to-end during training.

### 5.4. Classification with Image Inputs

To evaluate whether the fairness layer performs well with large-scale datasets and complex model architectures, we apply the method to two classification tasks with high-

dimensional image inputs. The first task uses the CelebA dataset (Liu et al., 2015), which contains over 200,000 images of faces, and the prediction task is to determine whether a person is smiling. This setting is motivated by smile detection in smartphone cameras, where systematic differences in prediction quality across demographic groups may lead some users to receive consistently worse auto-captured photos. The second task uses the FairFace dataset (Kärkkäinen & Joo, 2021), which contains over 100,000 face images with demographic annotations. The prediction task is to classify whether a person is over or under 30 years old. This task is motivated by age-gated access systems, such as content filtering or retail verification, where systematic prediction differences across demographic groups may incorrectly deny access to certain users.

For each dataset and fairness method, we train models across five image architectures using GPU acceleration. The models include ViT-B/16 (Kolesnikov et al., 2021) (a vision transformer with LoRA adapters (Hu et al., 2022)), Swin-T (Liu et al., 2021) (a shifted-window hierarchical transformer), DenseNet-121 (Huang et al., 2016) (a densely connected convolutional network), ResNet-18 (He et al., 2015) (a residual convolutional network), and a convolutional neural network trained from scratch. These architectures span both convolutional and transformer-based image models, enabling evaluation of whether the fairness layer remains effective across substantially different feature representations and optimization regimes.

*Table 2.* Image classification results for each fairness method and model architecture. All methods satisfied the imposed fairness constraints. The F-Layer method consistently achieves the highest accuracy, followed by the Projection method. The Penalty and Strict Penalty methods obtain similar accuracies as feasibility-prioritized cross-validation selected penalty weights close to the large values used in the Strict Penalty regime, causing both objectives to heavily prioritize constraint satisfaction (as is necessary in high-stakes regimes) over classification loss.

| Backbone | Method | CelebA Acc. | FairFace Acc. |
|---|---|---|---|
| Swin-T | F-Layer | **0.918** | **0.808** |
| | Projection | 0.888 | 0.765 |
| | Penalty | 0.501 | 0.551 |
| | Strict Penalty | 0.501 | 0.551 |
| ResNet-18 | F-Layer | **0.904** | **0.762** |
| | Projection | 0.885 | 0.748 |
| | Penalty | 0.500 | 0.550 |
| | Strict Penalty | 0.501 | 0.550 |
| ViT-B/16 | F-Layer | **0.906** | **0.775** |
| | Projection | 0.876 | 0.754 |
| | Penalty | 0.501 | 0.550 |
| | Strict Penalty | 0.501 | 0.550 |
| DenseNet-121 | F-Layer | **0.915** | **0.762** |
| | Projection | 0.893 | 0.739 |
| | Penalty | 0.501 | 0.551 |
| | Strict Penalty | 0.501 | 0.551 |
| CustomCNN | F-Layer | **0.814** | **0.673** |
| | Projection | 0.809 | 0.670 |
| | Penalty | 0.501 | 0.550 |
| | Strict Penalty | 0.501 | 0.550 |

#### 5.4.1. Constraints

In the CelebA dataset setting, we impose constraints requiring the average predicted logits of whether the person is smiling or not to be similar across race and gender groups, while in the FairFace dataset setting, we impose constraints requiring similar average predictions across intersectional race and gender groups for the age prediction task.

For both experiments, the fairness layer is applied to the scalar logit for the positive class and is defined as:

$$g(z) = \arg\min_{p} \|p - z\|_2^2 \quad \text{s.t.} \begin{cases} \Bigg| \frac{1}{n_b - \sum_i x_{ij}} \sum_{\{i \,:\, x_{ij}=0\}} p_i \\ \quad - \frac{1}{\sum_i x_{ij}} \sum_{\{i \,:\, x_{ij}=1\}} p_i \Bigg| \leq \epsilon_{\text{img}} \quad \forall j \in \mathcal{S}_{\text{img}}. \end{cases} \tag{12}$$

Here $\mathcal{S}_{\text{img}}$ denotes the set of binary demographic group indicators used in the image experiments, and $\epsilon_{\text{img}}$ ($1 \times 10^{-4}$ for FairFace and $1 \times 10^{-3}$ for CelebA) is the allowed tolerance in average predicted logits across groups. The F-Layer and Projection methods employ the projection defined in (12), while the Penalty and Strict Penalty baselines minimize the corresponding regularized objectives described in Appendix E.

#### 5.4.2. Results

Table 2 shows that the F-Layer method consistently improves accuracy over all baselines. Relative gains over the Projection method are typically 2–5%, while gains over Lagrangian methods are larger. See Appendix F for an ablation study on experiment hyperparameters.

### 5.5. Synthetic Data Setting

#### 5.5.1. Constraints

In the synthetic regression experiments, we use:

$$g(z) = \arg\min_{p} \|p - z\|_2^2 \quad \text{s.t.} \begin{cases} \Bigg| \frac{1}{n - \sum_i x_{ij}} \sum_{\{i \,:\, x_{ij}=0\}} p_i \\ \quad - \frac{1}{\sum_i x_{ij}} \sum_{\{i \,:\, x_{ij}=1\}} p_i \Bigg| \leq 0.05 \\ p_i \leq u \quad \forall i, \quad p_i \geq \ell \quad \forall i. \end{cases} \tag{13}$$

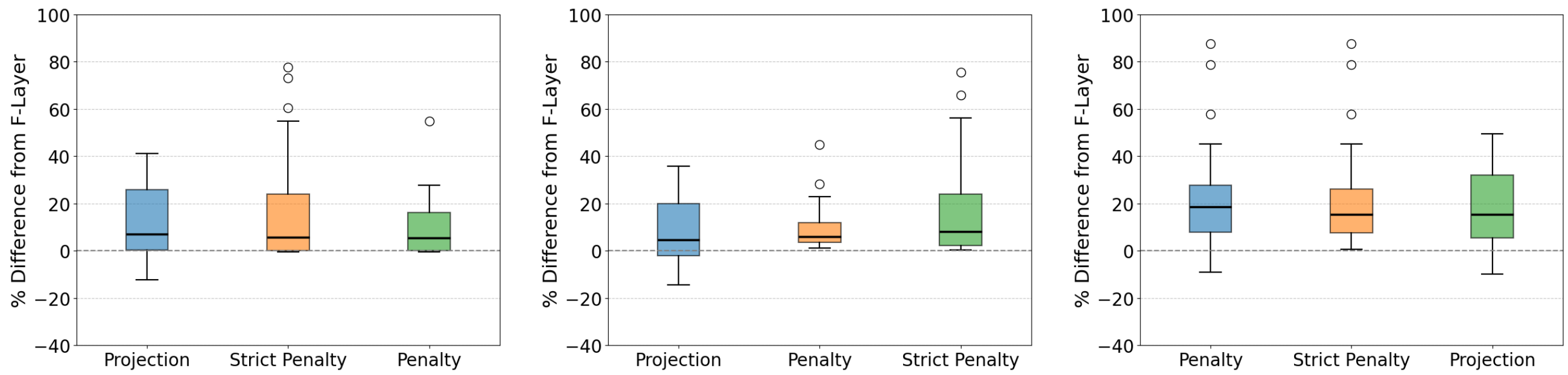

*(a)* $b_{\text{train}} = 2000, b_{\text{infer}} = 2000$ *(b)* $b_{\text{train}} = 2000, b_{\text{infer}} = 16$ *(c)* $b_{\text{train}} = 256, b_{\text{infer}} = 16$

*Figure 2.* Test loss percent differences relative to F-Layer across 32 datasets. Each boxplot shows $\{(L_i^{(j)} - F_i)/F_i \times 100\}_{i=1}^{32}$ where $F_i$ is F-Layer loss and $L_i^{(j)}$ is baseline $j$ loss. Penalty model excludes cases where fairness constraints were violated.

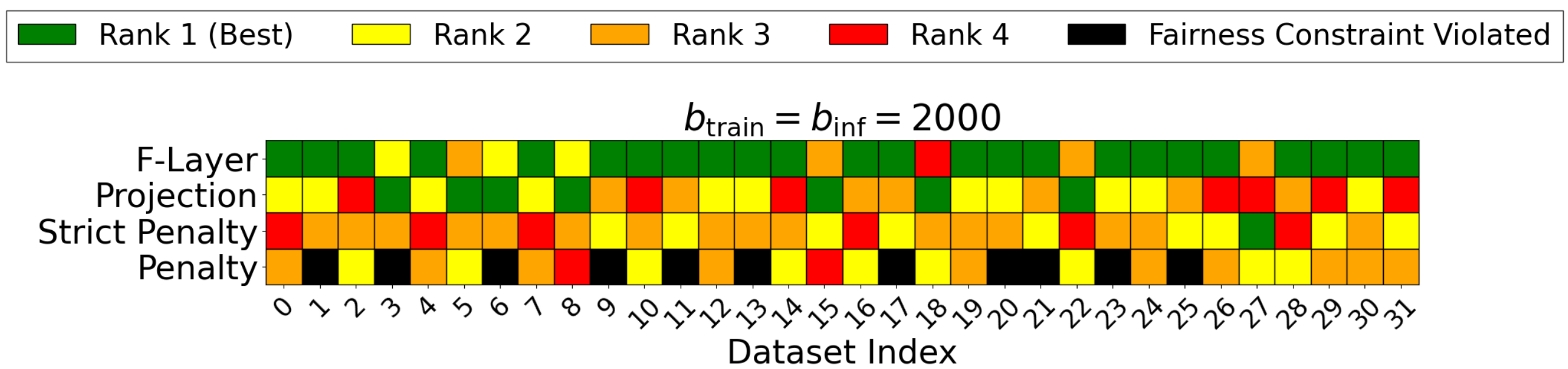

*Figure 3.* Test loss rankings across all 32 dataset scenarios.

to define the fairness layer. Here $z = f_\theta(X^{(b)}) \in \mathbb{R}^{n_b}$. In other words, the constraint set ensures predictions stay within some bounds and Mean Conditional Parity (2) is satisfied within a small tolerance $\epsilon = 0.05$. See Appendix E for the corresponding Penalty and Strict Penalty formulations.

The F-Layer, Projection, Penalty, and Strict Penalty models have the same architecture up to the final layer; all models have 15 hidden layers with ReLU activation functions, and layer normalization is performed per-layer.

### 5.5.2. Results

All models are trained via SGD with $b_{\text{train}} \in \{256, 2000\}$ using SGD ($\eta = 10^{-4}$, decay of 0.66 after 8 epochs without improvement, and early stopping with a patience of 25 epochs). Figure 2 shows test loss distributions relative to F-Layer, Figure 3 displays rankings across all datasets in the large batch regime, and Figure 4 in Appendix E.4 displays rankings across all scenarios. We highlight the key findings:

Consistency: F-Layer ranked first in 22–27 of 32 datasets across all batch size combinations (the range reflects variation across different $(b_{\text{train}}, b_{\text{inf}})$ configurations), demonstrating robustness to batch size regime and validating the online algorithm for small-batch inference.

Significant improvements: F-Layer achieved 18–30% test loss reductions on 16–24 of 32 datasets, depending on batch size configuration. Median improvements were approximately 8% vs all baseline methods when $b_{\text{train}} = 2000$ and 20% vs all baseline methods when $b_{\text{train}} = 256$.

Penalty failures: Penalty methods frequently violated constraints despite validation-based $\lambda$ selection. For $b_{\text{train}} = b_{\text{infer}} = 2000$, 7 of 11 violations had fairness gaps 7 times larger than desired, with 4 exceeding 15 times the target.

F-Layer limitations: In the large training and inference batch regime, F-Layer underperformed on 6–8 datasets with tight constraints ($\ell = 0, u = 3.5$), suggesting training difficulty in restricted feasible regions. Post-hoc projections may avoid local minima in such cases.

## 6. Conclusion

The "fairness layer" is a neural network component that guarantees fairness while maintaining end-to-end differentiability, and numerical experiments show consistent improvements over common in-processing and post-processing methods. Promising directions for future work include incorporating nonconvex constraints into differentiable optimization layers and providing theoretic guarantees of when differentiable optimization layers can outperform post-hoc projections onto a constraint set.

## Acknowledgements

This project has been supported in part by the DARPA AIQ grant HR00112520014. GM has been supported in part by NSF grants DMS-2522495, DMS-2145630, CCF-2212520, and by DFG SPP 2298 project 464109215, and the BMFTR in DAAD project 57616814 (SECAI).

## Impact Statement

This work introduces methods for enforcing fairness constraints in neural networks with mathematical guarantees. There are numerous societal implications stemming from this work that warrant important considerations:

*Potential Positive Impacts*: Our fairness layer provides verifiable guarantees that specified notions of fairness are satisfied during model deployment, which may help organizations comply with emerging AI regulations. By maintaining end-to-end differentiability, our method enables practitioners to incorporate fairness considerations directly into model training rather than as an afterthought, potentially leading to more trustworthy AI systems.

*Limitations and Potential for Misuse*: First, our work focuses exclusively on group fairness metrics, which capture only one aspect of existing fairness desiderata; moreover, definitions of fairness may conflict with one another. This work does not argue that certain fairness definitions are more useful or appropriate than others. Second, the choice of protected attributes, fairness constraints, and tolerance parameters requires careful consideration and domain expertise. Inappropriate choices could introduce new forms of unintended disparity.

*Broader Context*: We encourage practitioners to view our method as one tool within a comprehensive fairness strategy, not a complete solution to fairness in AI systems. They should complement and not replace broader efforts including transparent model documentation and ongoing monitoring. As discussed in Appendix C, differentiable layers can be utilized for a wide array of applications beyond fairness. In such applications, we encourage a similar strategy of model transparency and monitoring.

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

## A. Differentiation Through Fairness Layer

Critically, the Jacobian with respect to $z = f_\theta(X^{(b)})$ can be obtained by differentiating the KKT conditions of the optimization problem (Gould et al., 2021). In this setting, these computations enable backpropagation through the fairness layer. Let

$$\mathcal{L}(\tilde{y}, \lambda, \nu; z) = \tilde{d}(\tilde{y}, z) + \lambda^\top(A\tilde{y} - m_1) + \nu^\top(B\tilde{y} - m_2) \quad (14)$$

denote the Lagrangian function, and let $(\tilde{y}^\star, \lambda^\star, \nu^\star)$ denote a primal-dual optimal point. The KKT optimality conditions are:

$$\nabla_{\tilde{y}} d(\tilde{y}^\star, z) + A^\top \lambda^\star + B^\top \nu^\star = 0 \quad (15)$$
$$A\tilde{y}^\star - m_1 \leq 0 \quad (16)$$
$$B\tilde{y}^\star - m_2 = 0 \quad (17)$$
$$\lambda^\star \geq 0, \quad \lambda_i^\star (A\tilde{y}^\star - m_1)_i = 0 \quad \forall i \in [q] \quad (18)$$

Let $\mathcal{A} = \{i : \lambda_i^\star > 0\}$ denote the set of active inequality constraints. Linearizing the KKT system around $(\tilde{y}^\star, \lambda^\star, \nu^\star)$ by taking the total differential with respect to $z$ yields the block linear system:

$$\begin{bmatrix} \nabla^2_{\tilde{y}\tilde{y}} \tilde{d}(\tilde{y}^*, z) & A_{\mathcal{A}}^\top & B^\top \\ A_{\mathcal{A}} & 0 & 0 \\ B & 0 & 0 \end{bmatrix} \begin{bmatrix} \mathrm{d}\tilde{y} \\ \mathrm{d}\lambda_{\mathcal{A}} \\ \mathrm{d}\nu \end{bmatrix} = - \begin{bmatrix} \nabla^2_{\tilde{y}z} d(\tilde{y}^\star, z)\, \mathrm{d}z \\ 0 \\ 0 \end{bmatrix}, \quad (19)$$

where $A_{\mathcal{A}}$ contains only the rows of $A$ corresponding to active constraints and $\nabla^2_{\tilde{y}\tilde{y}}\mathcal{L} = \nabla^2_{\tilde{y}\tilde{y}}\tilde{d}(\tilde{y}^\star, z)$ is the Hessian of the dissimilarity function. Denoting the KKT matrix as $K$, by the implicit function theorem under the stated regularity conditions, $K$ is nonsingular, and thus $\mathrm{d}\tilde{y} = J_g(z)\,\mathrm{d}z$ is obtained by solving this linear system. In practice, auto-differentiation frameworks only require Jacobian-vector products, so backpropagation through $g$ reduces to solving a single linear system involving $K^\top$. This yields exact gradients of the projected predictions $\hat{y} = g(z)$ with respect to the raw network outputs $z = f_\theta(X^{(b)})$, enabling stable and fully end-to-end training.

## B. Nonconvex Fairness Constraints

In this work, we focus on notions of fairness of the form enumerated in (1). However, as opposed to expected values, some notions of group fairness in the classification setting involve the proportion of positive or negative classifications across protected classes; some instances include the most common form of demographic parity (Hardt et al., 2016) and equalized odds (Hardt et al., 2016). In the case of demographic parity, for example, consider a binary classification setting with 2 protected classes:

$$Pr(\tilde{y} = 1 | X_{:j} = 0) = Pr(\tilde{y} = 1 | X_{:j} = 1) \quad (20)$$

To practically enforce this notion of fairness during training with a differentiable optimization layer, one would define the fairness constraint as follows (using the same notation as in (13)):

$$\left| \frac{1}{n - \sum_i x_{ij}} \sum_{\{i \,:\, x_{ij}=0\}} \mathbf{1}_{p_i \geq 0.5} - \frac{1}{\sum_i x_{ij}} \sum_{\{i \,:\, x_{ij}=1\}} \mathbf{1}_{p_i \geq 0.5} \right| \leq \varepsilon \quad (21)$$

Here $\mathbf{1}_{p_i \geq 0.5}$ is an indicator function equaling 1 if the predicted probability is greater than 0.5 and 0 otherwise. This indicator function breaks both the convex and differentiable nature of the general fairness layer definition (1), thereby becoming computationally impractical for many applications.

To attempt to overcome the nonconvex nature of such fairness constraints, a number of solution methods have been explored. For example, convex surrogates can approximate the proportion of positive predictions for a given class, thus ensuring a proxy of demographic parity (Wu et al., 2019). Similarly, convex-concave (difference of convex functions) formulations have been applied to ensure classifiers with convex decision boundaries satisfy other fairness metrics (Zafar et al., 2017a;b). Such methods either ensure an approximation of the original fairness constraints are satisfied or do not apply to the setting of neural networks. In contrast, the focus of this work is to guarantee notions of fairness when training and employing neural network model architectures for inference.

## C. Extensions Beyond Fairness

While we instantiate the linear constraints in (1) for the purpose of fairness, there are other clear applications that fit within this framework. We provide 3 such examples:

*Portfolio Optimization.* Neural models are increasingly being utilized to make investment decisions in portfolio optimization settings. Such decisions must adhere to natural budgetary or risk management constraints. The problem can be formulated as a convex quadratic program:

$$\begin{aligned} \hat{x} = \arg\min_x \quad & \frac{1}{2}\|x - z\|_2^2 \\ \text{s.t.} \quad & \mathbf{1}^\top x = 1 \\ & x \geq 0, \\ & x \leq u_{\max} \\ & Sx \leq s_{\max} \end{aligned}$$

Where $z \in \mathbb{R}^n$ is the original network's predicted investment allocation vector, $u_{\max}$ is a vector of maximum al-

lowable allocations per asset, $S$ is a matrix encoding sector membership (i.e. rows represent various sectors and columns represent the investment assets), $s_{\max}$ limits the total exposure to each sector, and $\hat{x}$ is the feasible projected allocation satisfying all constraints.

*Resource Allocation and Task Assignment.* Neural models can be used to predict resource assignments to tasks; for example, the model may predict required CPU cores, bandwidth, or personnel for a task. To ensure that capacity limits are not violated, a differentiable optimization layer can adjust these outputs:

$$\hat{r} = \arg\min_{r} \|r - z\|_2^2 \quad \text{s.t.} \quad Ar \leq b, \quad r \geq 0.$$

Where $r$ is the resource assignment vector, $z$ is the unconstrained network prediction, and $Ar \leq b$ represents linear capacity constraints. The matrix $A$ encodes which resources share a common capacity limit; each row of $A$ corresponds to a linear constraint (e.g., a machine or network link), and each column corresponds to a task or resource variable. For instance, suppose we have four tasks $r_1, r_2, r_3, r_4$ and two shared resources (machines or network links). Then a possible assignment could be represented as:

$$r = \begin{bmatrix} r_1 \\ r_2 \\ r_3 \\ r_4 \end{bmatrix}, \quad A = \begin{bmatrix} 1 & 0 & 1 & 0 \\ 0 & 1 & 1 & 1 \end{bmatrix}, \quad b = \begin{bmatrix} b_1 \\ b_2 \end{bmatrix}.$$

The resulting linear constraint is $Ar \leq b$, which ensures that each shared resource's total usage does not exceed its capacity. The 1's in each row indicate which tasks contribute to that resource, while 0's indicate no contribution.

*Robotics and Autonomous Systems.* A neural network may predict candidate control inputs or trajectories; a trajectory is a planned sequence of positions, velocities, or actions over time that the robot should follow to achieve a task. These outputs often need to respect safety or physical constraints, such as velocity, acceleration, or joint limits:

$$\hat{u} = \arg\min_{u} \|u - z\|_2^2 \quad \text{s.t.} \quad Cu \leq d,$$

where $u$ represents control inputs, $z$ is the unconstrained network prediction, and $Cu \leq d$ encodes linear safety or actuation limits. Specifically, each row in the matrix $C$ represents a linear constraint, and each column represents a control variable. Control variables can include actuator inputs (commands sent to motors to move robot limbs), joint angles or torques (the positions, velocities, or torques at robot joints that determine how the robot moves), or velocities and accelerations (desired linear or angular speeds of parts of the robot or vehicle along the trajectory).

# D. Datasets in Numerical Experiments

## D.1. Loan Default Dataset

We use a public dataset from the United States Small Business Administration (SBA) (Mickle, 2019) to train the F-Layer and baseline models to predict whether a business applying for a loan will eventually default. The dataset contains 2102 observations of applications submitted for approval with 35 features such as the amount dispersed, length of loan term, whether the business is new, and the originating bank. The target variable is a binary outcome indicating whether the business defaulted on the loan. We one-hot encode all categorical variables, standardize numeric features, and split data into training, validation, and test sets. For models that minimize a loss function that includes a penalty term for constraint violations, we use employ the validation set to choose $\lambda$ via cross validation.

## D.2. Employee Performance Dataset

The dataset contains 1200 observations of employees with over 25 features such as employee age, education level, internal performance rating, average customer satisfaction score, and hourly wage (Gupta, 2023). We one-hot encode all categorical variables–leading to 62 total predictors–and standardize both the training data and target variable.

## D.3. Synthetic Datasets

Numerical experiments are performed across 32 synthetically generated datasets for a typical predictive inference task. Each dataset has $n$ = 40,000 observations and $d = 150$ features. A 70%/6%/24% training/validation/test set split, respectively, is used. To holistically compare the fairness methods, datasets and fairness layers were created with a variation of key properties. Table 3 details these properties.

To create each dataset, a binary protected attribute $X_{:j} \in \{0,1\}^n$ is first sampled from a Bernoulli distribution with success probability $p \in \{0.2, 0.5\}$, determining the degree of class imbalance. Conditional on this attribute, feature vectors $x \in \mathbb{R}^d$ are generated as correlated Gaussian random variables. Specifically, the feature matrix is divided into correlation blocks of size approximately 20, with within-block covariance parameterized by a base correlation $\rho = 0.3$. For each block $B$, features are sampled independently from a sample normal distribution. Then, given the Cholesky factorization of the desired covariance matrix of the block $Cov(B) = LL^\top$, we multiply $X_{\text{block}} = BL^\top$ to obtain the final feature block with the desired correlation structure. After obtaining these blocks, a subset of 50 features is then perturbed to be linearly correlated with the protected attribute, where the strength of the correlation is determined by the "group relevance" parameter (0.3 or 0.7). This in-

*Table 3.* The various dataset and fairness layer properties explored in the synthetic regression experiments. The first 5 properties were combined to create $2^5 = 32$ different datasets for the synthetic regression setting. Each model was tested on each dataset 3 times: once with $b_{\text{train}} = b_{\text{infer}} = 2000$, once with $b_{\text{train}} = b_{\text{infer}} = 20$, and once with $b_{\text{train}} = 2000$, $b_{\text{infer}} = 20$.

| NAME | DESCRIPTION | VALUES |
|---|---|---|
| CLASS IMBALANCE | PERCENT OF OBSERVATIONS WITH PROTECTED ATTRIBUTE VALUE = 1 | 20%, 50% |
| GROUP RELEVANCE | (1) CORRELATION BETWEEN GROUP LABEL AND INPUT FEATURES (2) GROUP BIAS ADDED TO TRUE SCORES | (1) 0.3, 0.7 (2) $\pm 3$, $\pm 6$ |
| NOISE LEVEL | STD. DEV. OF GAUSSIAN NOISE ADDED TO TARGETS | 0.125, 0.6 |
| CONSTRAINT TIGHTNESS | VALUES FOR $\ell$ AND $u$ IN (13) | "LOOSE": $\ell = -3.5, u = 3.5$
"TIGHTER": $\ell = 0, u = 3.5$ |
| DATA STRUCTURE | HOW TARGET VALUES WERE GENERATED | LINEAR, NONLINEAR |
| $b_{\text{TRAIN}}$ | BATCH SIZE DURING TRAINING | 2000, 20 |
| $b_{\text{INFER}}$ | BATCH SIZE DURING INFERENCE | 2000, 20 |

duces systematic differences between the two groups in both marginal and conditional distributions.

Targets $y$ are generated from either a linear or nonlinear data-generating process. Both methods generate a latent score $s$ and share the following structure when generating final target observations $y^*$:

$$y^* = s + b + \epsilon, \quad \epsilon \sim N(0, \sigma) \text{ and}$$
$$b_i = \begin{cases} +b, & x_{ij} = 0 \text{ for } j \in \mathcal{S}, \\ -b, & x_{ij} = 1 \text{ for } j \in \mathcal{S}. \end{cases}$$

The methods generate the latent score $s$ in different ways. For the linear method, a sparse coefficient vector $\beta \in \mathbb{R}^d$ with nonzero entries on 15 randomly selected features is used to compute the latent score $s = x^\top \beta$, where nonzero entries of $\beta$ are sampled from a standard normal distribution.

For the nonlinear datasets, the target includes higher-order terms, combining polynomial transformations and random pairwise interactions among features:

$$s = 0.7(x^\top \beta) + 0.2 \sum_j \alpha_j x_j^2 + 0.1 \sum_{k \neq l} \gamma_{kl} x_k x_l, \quad (22)$$

where coefficients $\alpha_j \sim N(0, .4)$, $\gamma_{kl} \sim N(0, .4)$.

After generating targets $y^*$, all values are normalized to have zero mean and unit variance before training, and the variance of observations with protected attribute value 0 is slightly reduced by scaling its values by 0.85 to simulate unequal outcome variability across groups.

# E. Regularized Loss Functions in Numerical Experiments

## E.1. Loan Default Experiments

The Strict Penalty method minimizes the following regularized loss function:

$$\mathcal{L}_{\text{penalty}} = -\frac{1}{N} \sum_{i=1}^{N} [y_i \log(\sigma(z_i)) + (1 - y_i) \log(1 - \sigma(z_i))]$$
$$+ \lambda \sum_{j \in \mathcal{S}} \left( \frac{1}{n - \sum_i x_{ij}} \sum_{\{i \,:\, x_{ij}=0\}} z_i - \frac{1}{\sum_i x_{ij}} \sum_{\{i \,:\, x_{ij}=1\}} z_i \right)^2$$

When performing cross validation for all 25 data splits to find suitable values of $\lambda$, we find that the optimization landscape is extremely sensitive to different weighting values. For example, the degree of constraint violation between the validation and test sets often varied widely, and increasing $\lambda$ by small amounts often led to the model predicting either every observation defaulting or every observation not defaulting on the loans. This pattern persisted regardless of the chosen form of the penalty term (quadratic, absolute value, etc.). Due to the difficulty of finding values of $\lambda$ that lead to constraint satisfaction without degenerate model behavior, we only employ the Strict Penalty model with $\lambda = 1000$ and do not include the Penalty model in calculations. However, the Penalty model is included in comparisons for other data settings (Section 5.3 and Section 5.5).

## E.2. Employee Performance Experiments

In addition to the F-Layer method and Projection method with projection defined by (11), we utilize the following

augmented loss function for the Penalty method:

$$\mathcal{L}_{\text{penalty}} = \frac{1}{N}\sum_{i=1}^{N}\|y_i-\hat{y}_i\|_2^2 + \lambda\sum_{j\in\mathcal{S}}\left(\left|\frac{1}{n-\sum_i x_{ij}}\sum_{\{i\,:\,x_{ij}=0\}}\hat{y}_i-y_i\right| + \left|\frac{1}{\sum_i x_{ij}}\sum_{\{i\,:\,x_{ij}=1\}}\hat{y}_i-y_i\right|\right). \tag{23}$$

Here $\lambda$ is the penalty weight controlling the trade-off between prediction accuracy and fairness. We re-train the model across a grid of candidate $\lambda$ values and pick the smallest $\lambda$ such that the fairness constraints are satisfied.

### E.3. Image Classification Experiments

In addition to the F-Layer method and Projection method with projection defined by (12), the Penalty and Strict Penalty methods minimize an augmented binary cross-entropy loss with a squared pairwise group-gap penalty. Let $s_i$ denote the demographic group label for observation $i$, let $\mathcal{G}$ denote the set of demographic groups appearing in the batch, and define $I_g = \{i : s_i = g\}$. The augmented objective is then:

$$\mathcal{L}_{\text{penalty}} = -\frac{1}{N}\sum_{i=1}^{N}\left[y_i\log(\sigma(z_i)) + (1-y_i)\log(1-\sigma(z_i))\right] + \lambda\sum_{\substack{g,h\in\mathcal{G}\\ g<h}}\left(\frac{1}{|I_g|}\sum_{i\in I_g}z_i - \frac{1}{|I_h|}\sum_{i\in I_h}z_i\right)^2. \tag{24}$$

Here, $z_i$ is the raw logit output of the model for sample $i$, $\sigma(\cdot)$ denotes the sigmoid function, and $\lambda > 0$ controls the strength of the fairness penalty. The penalty term is the sum of squared pairwise differences between average logits across demographic groups in the batch. As in the other experiments, the Penalty method selects $\lambda$ from a candidate set using validation performance subject to constraint satisfaction, while the Strict Penalty method uses a large fixed value of $\lambda$ to strongly encourage constraint satisfaction.

### E.4. Synthetic Experiments

The Penalty and Strict Penalty methods minimize the following augmented loss:

$$\mathcal{L}_{\text{penalty}} = \frac{1}{N}\sum_{i=1}^{N}\|y_i-\hat{y}_i\|_2^2 + \lambda\left|\frac{1}{n-\sum_i x_{ij}}\sum_{\{i\,:\,x_{ij}=0\}}\hat{y}_i - \frac{1}{\sum_i x_{ij}}\sum_{\{i\,:\,x_{ij}=1\}}\hat{y}_i\right|. \tag{25}$$

*Table 4.* Image classification accuracy under the larger-slack setting.

| Backbone | Method | CelebA | FairFace |
|---|---|---|---|
| ViT-B/16 | F-Layer | **0.907** | **0.777** |
| | Projection | 0.861 | 0.748 |
| | Penalty | 0.522 | 0.551 |
| DenseNet | F-Layer | **0.898** | **0.739** |
| | Projection | 0.874 | **0.739** |
| | Penalty | 0.501 | 0.551 |
| Swin-T | F-Layer | **0.895** | **0.774** |
| | Projection | 0.871 | 0.765 |
| | Penalty | 0.645 | 0.551 |
| ResNet-18 | F-Layer | **0.904** | **0.765** |
| | Projection | 0.870 | 0.743 |
| | Penalty | 0.705 | 0.551 |
| CustomCNN | F-Layer | **0.815** | **0.670** |
| | Projection | 0.794 | 0.663 |
| | Penalty | 0.501 | 0.551 |

To incorporate the box constraints from (13) in the Penalty and Strict Penalty methods, the outputs from the model $z = f_\theta(X)$ are transformed via the shifting function $\hat{y} = \ell + (u-\ell)\sigma(z)$. Whereas the Strict Penalty method always uses a large penalty $\lambda = 5000$ for all datasets, the Penalty model chooses $\lambda$ from a candidate set via cross validaton.

## F. Image Classification Experiments Ablation Study

**Ablation 1: Constraint Tightness Threshold ($\epsilon$).** To ensure that the original image classification results are not simply a byproduct of a specific constraint tightness, we relax the allowed slack from $10^{-3}$ to $10^{-2}$ for the CelebA experiments and from $10^{-4}$ to $10^{-3}$ for the FairFace experiments. Overall, as shown in Table 4, we observe the same pattern as in the original experiments. The F-Layer method consistently improves accuracy over the Projection method, with gains ranging from approximately 1% to 5% depending on the architecture and dataset. If the unconstrained and constrained solutions are close in distance (i.e. constraint slack is extremely loose), one would expect F-Layer and Projection methods to achieve more similar results as neither projection would alter the unconstrained solution.

**Ablation 2: Primal-Dual Threshold ($b_\tau$).** We next ablate the primal-dual threshold $b_\tau$ used by the inference algorithm. The results in Tables 5 and 6 show that the choice of $b_\tau$ has negligible impact on accuracy for moderate inference batch sizes. For example, for a fixed model and dataset, the rows corresponding to $b_\tau = 4$ and $b_\tau = 1024$ are nearly identical once $b_{\text{infer}}$ is moderately large. At very small batch sizes, particularly when the batch size is comparable to the number of protected groups, the primal-dual regime can

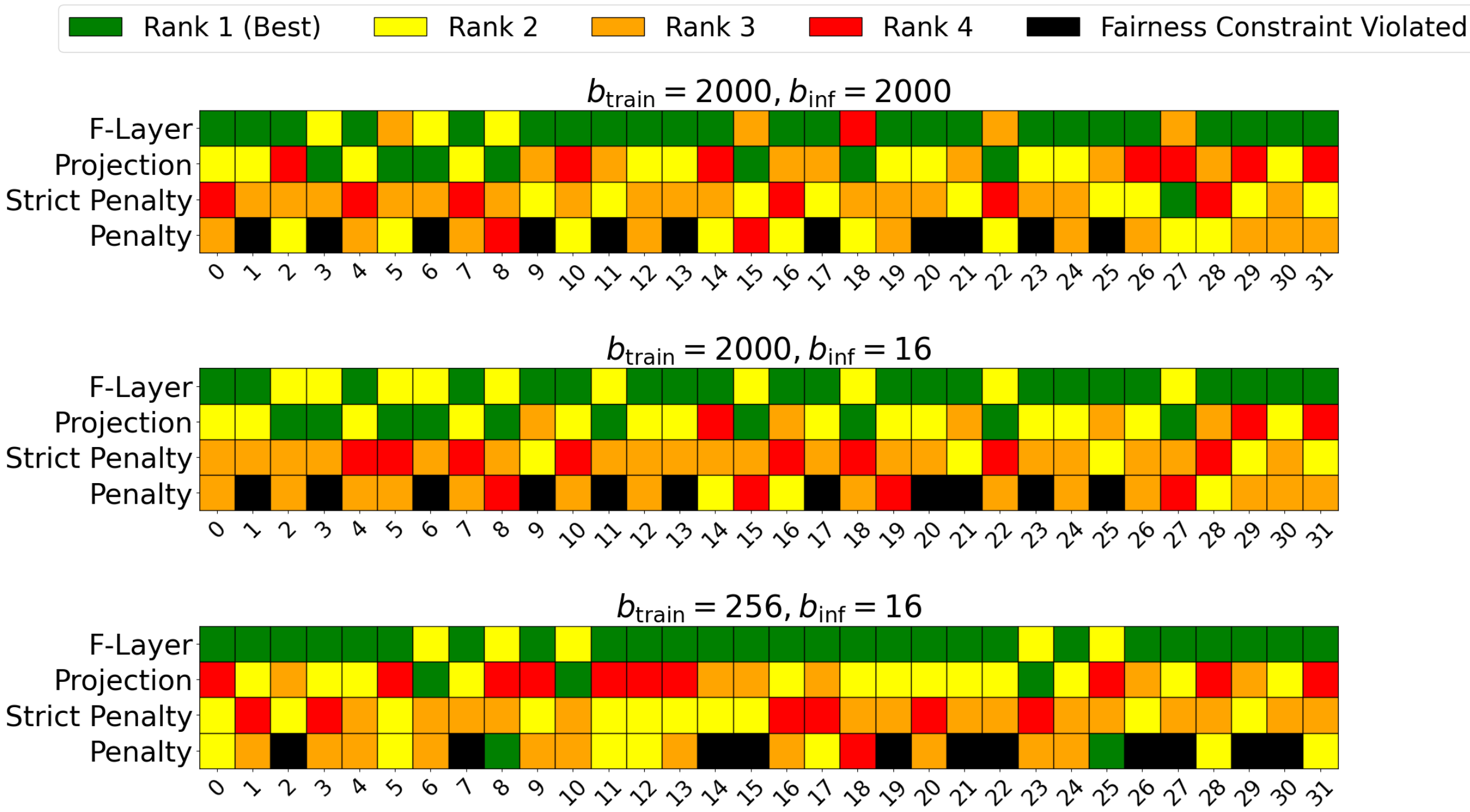

*Figure 4.* Test loss rankings across all 32 dataset scenarios and batch size combinations. Black cells indicate constraint violations.

yield modest accuracy gains over hard projection, consistent with softer enforcement allowing individual batches more expressivity.

When stratified sampling is used, as in this ablation study, both regimes achieve aggregate fairness by Lemma 3.1. In streaming settings where stratified sampling is infeasible, Theorem 3.2 guarantees that the sample-weighted average per-batch fairness violation still converges to at most $\epsilon$.

## G. Comparison of Fairness Methods

Table 7 provides a comparison between existing fairness methods in relation to goals (G1), (G2), and (G3) enumerated in Section 1.3.

## H. Proofs

### H.1. Proof of Lemma 3.1

*Proof.* We decompose the aggregate fairness gap into a baseline term (assuming equal group proportions across batches) plus correction terms capturing the effect of varying proportions. Let $n_b = n_{b,0} + n_{b,1}$ denote the size of batch $b$, and let $N = \sum_{b=1}^{B} n_b$ denote the total number of samples across all batches.

We can express the aggregate statistics in terms of batch proportions:

$$\mathcal{F}_0 = \frac{\sum_{b=1}^{B} n_{b,0}\mathcal{F}_{b,0}}{\sum_{b=1}^{B} n_{b,0}} = \frac{\sum_{b=1}^{B} p_b n_b \mathcal{F}_{b,0}}{\bar{p}N} \tag{26}$$

$$\mathcal{F}_1 = \frac{\sum_{b=1}^{B} n_{b,1}\mathcal{F}_{b,1}}{\sum_{b=1}^{B} n_{b,1}} = \frac{\sum_{b=1}^{B} (1-p_b) n_b \mathcal{F}_{b,1}}{(1-\bar{p})N} \tag{27}$$

**Step 1: Equal proportions baseline.** Consider the hypothetical aggregate statistics if all batches had the same group proportion $\bar{p}$:

$$\tilde{\mathcal{F}}_0 = \frac{\sum_{b=1}^{B} n_b \mathcal{F}_{b,0}}{\sum_{b=1}^{B} n_b} = \frac{1}{N}\sum_{b=1}^{B} n_b \mathcal{F}_{b,0} \tag{28}$$

$$\tilde{\mathcal{F}}_1 = \frac{\sum_{b=1}^{B} n_b \mathcal{F}_{b,1}}{\sum_{b=1}^{B} n_b} = \frac{1}{N}\sum_{b=1}^{B} n_b \mathcal{F}_{b,1} \tag{29}$$

*Table 5.* FairFace accuracy across inference batch sizes and primal-dual thresholds. The smallest batch size, $b_{\text{infer}} = 14$, equals the number of protected groups.

| **Backbone** | $b_\tau$ | $b_{\text{infer}} = 14$ | $b_{\text{infer}} = 16$ | $b_{\text{infer}} = 32$ | $b_{\text{infer}} = 64$ | $b_{\text{infer}} = 128$ | $b_{\text{infer}} = 256$ |
|---|---|---|---|---|---|---|---|
| ResNet-18 | 4 | 0.594 | 0.615 | 0.704 | 0.748 | 0.765 | 0.769 |
| | 1024 | 0.576 | 0.607 | 0.700 | 0.748 | 0.765 | 0.769 |
| ViT-B/16 | 4 | 0.587 | 0.606 | 0.717 | 0.758 | 0.777 | 0.784 |
| | 1024 | 0.579 | 0.607 | 0.717 | 0.758 | 0.777 | 0.784 |
| CustomCNN | 4 | 0.571 | 0.577 | 0.648 | 0.668 | 0.670 | 0.680 |
| | 1024 | 0.569 | 0.577 | 0.648 | 0.668 | 0.670 | 0.680 |

*Table 6.* CelebA accuracy across inference batch sizes and primal-dual thresholds.

| **Backbone** | $b_\tau$ | $b_{\text{infer}} = 4$ | $b_{\text{infer}} = 8$ | $b_{\text{infer}} = 16$ | $b_{\text{infer}} = 32$ | $b_{\text{infer}} = 64$ | $b_{\text{infer}} = 128$ |
|---|---|---|---|---|---|---|---|
| ResNet-18 | 4 | 0.765 | 0.778 | 0.827 | 0.862 | 0.877 | 0.889 |
| | 1024 | 0.751 | 0.775 | 0.825 | 0.862 | 0.877 | 0.889 |
| ViT-B/16 | 4 | 0.767 | 0.783 | 0.832 | 0.865 | 0.880 | 0.894 |
| | 1024 | 0.752 | 0.781 | 0.831 | 0.865 | 0.880 | 0.893 |
| CustomCNN | 4 | 0.705 | 0.714 | 0.754 | 0.781 | 0.789 | 0.801 |
| | 1024 | 0.702 | 0.714 | 0.753 | 0.781 | 0.788 | 0.801 |

We bound $|\tilde{\mathcal{F}}_0 - \tilde{\mathcal{F}}_1|$ as follows:

$$\begin{aligned}
|\tilde{\mathcal{F}}_0 - \tilde{\mathcal{F}}_1| &= \left| \frac{1}{N} \sum_{b=1}^{B} n_b \mathcal{F}_{b,0} - \frac{1}{N} \sum_{b=1}^{B} n_b \mathcal{F}_{b,1} \right| \\
&= \frac{1}{N} \left| \sum_{b=1}^{B} n_b (\mathcal{F}_{b,0} - \mathcal{F}_{b,1}) \right| \\
&\leq \frac{1}{N} \sum_{b=1}^{B} n_b |\mathcal{F}_{b,0} - \mathcal{F}_{b,1}| \\
&\leq \frac{1}{N} \sum_{b=1}^{B} n_b \epsilon = \epsilon
\end{aligned}$$

where the last inequality uses the per-batch fairness constraint $|\mathcal{F}_{b,0} - \mathcal{F}_{b,1}| \leq \epsilon$.

**Step 2: Deviation from equal proportions.** We now bound how much the actual aggregate statistics $\mathcal{F}_0$ and $\mathcal{F}_1$ deviate from the equal-proportions baseline $\tilde{\mathcal{F}}_0$ and $\tilde{\mathcal{F}}_1$:

$$\begin{aligned}
|\mathcal{F}_0 - \tilde{\mathcal{F}}_0| &= \left| \frac{\sum_{b=1}^{B} p_b n_b \mathcal{F}_{b,0}}{\bar{p} N} - \frac{\sum_{b=1}^{B} n_b \mathcal{F}_{b,0}}{N} \right| \\
&= \left| \frac{\sum_{b=1}^{B} (p_b - \bar{p}) n_b \mathcal{F}_{b,0}}{\bar{p} N} \right| \\
&\leq \frac{\sum_{b=1}^{B} |p_b - \bar{p}| \cdot n_b \cdot |\mathcal{F}_{b,0}|}{\bar{p} N} \\
&\leq \frac{\Delta_p \cdot R \sum_{b=1}^{B} n_b}{\bar{p} N} \\
&= \frac{\Delta_p \cdot R}{\bar{p}}
\end{aligned}$$

Similarly, we have:

$$\begin{aligned}
|\mathcal{F}_1 - \tilde{\mathcal{F}}_1| &= \left| \frac{\sum_{b=1}^{B} (1 - p_b) n_b \mathcal{F}_{b,1}}{(1 - \bar{p}) N} - \frac{\sum_{b=1}^{B} n_b \mathcal{F}_{b,1}}{N} \right| \\
&= \left| \frac{\sum_{b=1}^{B} (\bar{p} - p_b) n_b \mathcal{F}_{b,1}}{(1 - \bar{p}) N} \right| \\
&\leq \frac{\sum_{b=1}^{B} |p_b - \bar{p}| \cdot n_b \cdot |\mathcal{F}_{b,1}|}{(1 - \bar{p}) N} \\
&\leq \frac{\Delta_p \cdot R}{1 - \bar{p}}
\end{aligned}$$

**Step 3: Triangle inequality.** Combining the results from Steps 1 and 2 via the triangle inequality:

$$\begin{aligned}
|\mathcal{F}_0 - \mathcal{F}_1| &\leq |\mathcal{F}_0 - \tilde{\mathcal{F}}_0| + |\tilde{\mathcal{F}}_0 - \tilde{\mathcal{F}}_1| + |\tilde{\mathcal{F}}_1 - \mathcal{F}_1| \\
&\leq \frac{\Delta_p \cdot R}{\bar{p}} + \epsilon + \frac{\Delta_p \cdot R}{1 - \bar{p}} \\
&= \epsilon + \Delta_p \cdot R \left( \frac{1}{\bar{p}} + \frac{1}{1 - \bar{p}} \right)
\end{aligned}$$

□

### H.2. Proof of Theorem 3.2

*Proof.* We establish the aggregate fairness guarantee by analyzing the growth of the dual variable under diminishing step sizes and connecting it to cumulative constraint violations.

**Step 1: Bound growth of dual variable**

From the dual update, $\lambda_{t+1} = \max\{0, \lambda_t + \eta_t w_t(\hat{y}_{b_t})\}$ with

*Table 7.* Comparison of fairness methods for machine learning with respect to stated goals (G1–G3) described in Section 1.3.

| Method Category | Type | (G1) | (G2) | (G3) |
|---|---|---|---|---|
| **Fairness Layer (ours)** | In | ✓ | ✓ | ✓ |
| Post-hoc projection (Wei et al., 2020; Kim et al., 2019; Hardt et al., 2016) | Post | ✓ | ✗ | ✓ |
| Data reweighting / augmentation (Kamiran & Calders, 2012; Ktena et al., 2024) | Pre | ✗ | ✓ | ✓ |
| Lagrangian penalty (Padala & Gujar, 2021; Cotter et al., 2019; Celis et al., 2019) | In | ✗ | ✓ | ✓ |
| Convex surrogates / DCP (Zafar et al., 2017c;a; Wu et al., 2019) | In | ○[a] | ✓ | ✗[b] |

[a] Only enforces surrogate constraints of hard-thresholded problems, but theoretically can exactly satisfy affine constraints.
[b] Restricted to models with convex decision boundaries.

$\eta_t = \frac{\eta}{\sqrt{t}}$, the projection onto $\mathbb{R}_+$ ensures that

$$\lambda_{t+1}^2 \leq (\lambda_t + \eta_t w_t(\hat{y}_{b_t}))^2 = \lambda_t^2 + 2\eta_t \lambda_t w_t(\hat{y}_{b_t}) + \eta_t^2 w_t(\hat{y}_{b_t})^2 \tag{30}$$

$$\implies 2\eta_t \lambda_t w_t(\hat{y}_{b_t}) \geq \lambda_{t+1}^2 - \lambda_t^2 - \eta_t^2 w_t(\hat{y}_{b_t})^2 \tag{31}$$

Summing from $t = 1$ to $T$ and using $\lambda_1 = 0$:

$$2\sum_{t=1}^{T} \eta_t \lambda_t w_t(\hat{y}_{b_t}) \geq \lambda_{T+1}^2 - \sum_{t=1}^{T} \eta_t^2 w_t(\hat{y}_{b_t})^2 \tag{32}$$

By optimality of the primal update (6), for $\tilde{y}_{b_t}$ satisfying $w_t(\tilde{y}_{b_t}) \leq 0$ (by the assumption):

$$\|\hat{y}_{b_t} - \hat{y}_{\text{raw},t}\|^2 + \lambda_t w_t(\hat{y}_{b_t}) \leq \|\tilde{y}_{b_t} - \hat{y}_{\text{raw},t}\|^2 + \lambda_t w_t(\tilde{y}_{b_t}) \leq \|\tilde{y}_{b_t} - \hat{y}_{\text{raw},t}\|^2 \tag{33}$$

Since $\|\hat{y}_{b_t} - \hat{y}_{\text{raw},t}\|^2 \geq 0$, we obtain:

$$\lambda_t w_t(\hat{y}_{b_t}) \leq \|\tilde{y}_{b_t} - \hat{y}_{\text{raw},t}\|^2 \leq D^2, \tag{34}$$

where $D := \sup_t \|\tilde{y}_{b_t} - \hat{y}_{\text{raw},t}\| < \infty$ (predictions are bounded in practice).

Summing (34):

$$\sum_{t=1}^{T} \lambda_t w_t(\hat{y}_{b_t}) \leq D^2 T \tag{35}$$

For fairness constraints with bounded predictions the fairness gap satisfies $|v_t(\hat{y}_{b_t})| \leq V$ for some constant $V > 0$. Therefore:

$$|w_t(\hat{y}_{b_t})| = |b_t| \cdot |v_t(\hat{y}_{b_t}) - \epsilon| \leq |b_t|(V + \epsilon) \leq B(V + \epsilon), \tag{36}$$

where $B := \sup_t |b_t| < \infty$ is the maximum batch size.

Using (36):

$$\sum_{t=1}^{T} \eta_t^2 w_t(\hat{y}_{b_t})^2 \leq B^2(V + \epsilon)^2 \sum_{t=1}^{T} \frac{\eta^2}{t} = \eta^2 B^2 (V + \epsilon)^2 \sum_{t=1}^{T} \frac{1}{t} = O(\eta^2 B^2 \log T) \tag{37}$$

Combining (32), (35), and (37):

$$\lambda_{T+1}^2 \leq 2\sum_{t=1}^{T} \eta_t \lambda_t w_t(\hat{y}_{b_t}) + O(\eta^2 B^2 \log T) \leq 2D^2 \sum_{t=1}^{T} \eta_t + O(\eta^2 B^2 \log T) = 2D^2 \eta \sum_{t=1}^{T} \frac{1}{\sqrt{t}} + O(\eta^2 B^2 \log T) \tag{38}$$

Noting $\sum_{t=1}^{T} \frac{1}{\sqrt{t}} = 2\sqrt{T} + O(1)$, then:

$$\lambda_{T+1}^2 \leq 4D^2 \eta \sqrt{T} + O(\eta^2 B^2 \log T + D^2 \eta) = O(D^2 \eta \sqrt{T}), \tag{39}$$

where the second term is absorbed since $\log T = o(\sqrt{T})$.

Therefore, since the square root operator preserves asymptotic order:

$$\lambda_{T+1} = O(D\sqrt{\eta}\, T^{1/4}) \tag{40}$$

**Step 2: Cumulative violation bound**

The dual projection ensures that $\lambda_{t+1} = \max\{0, \lambda_t + \eta_t w_t(\hat{y}_{b_t})\} \geq \lambda_t + \eta_t w_t(\hat{y}_{b_t})$ holds for all $t$. This yields:

$$w_t(\hat{y}_{b_t}) \leq \frac{\lambda_{t+1} - \lambda_t}{\eta_t} = \frac{\sqrt{t}}{\eta}(\lambda_{t+1} - \lambda_t) \tag{41}$$

Summing from $t = 1$ to $T$:

$$\sum_{t=1}^{T} w_t(\hat{y}_{b_t}) \leq \frac{1}{\eta} \sum_{t=1}^{T} \sqrt{t}(\lambda_{t+1} - \lambda_t) \tag{42}$$

Applying summation by parts:

$$\sum_{t=1}^{T} \sqrt{t}(\lambda_{t+1} - \lambda_t) = \sqrt{T}\lambda_{T+1} - \sqrt{1}\lambda_1 - \sum_{t=1}^{T-1} \lambda_{t+1}(\sqrt{t+1} - \sqrt{t}) = \sqrt{T}\lambda_{T+1} - \sum_{t=1}^{T-1} \lambda_{t+1}(\sqrt{t+1} - \sqrt{t}) \tag{43}$$

since $\lambda_1 = 0$.

Since $\lambda_{t+1} \geq 0$ and $\sqrt{t+1} - \sqrt{t} > 0$ for all $t$, the sum $\sum_{t=1}^{T-1} \lambda_{t+1}(\sqrt{t+1} - \sqrt{t}) \geq 0$ is non-negative. Therefore:

$$\begin{aligned}\sum_{t=1}^{T} w_t(\hat{y}_{b_t}) &\leq \frac{\sqrt{T}\lambda_{T+1}}{\eta} \\ &= \frac{\sqrt{T} \cdot O(D\sqrt{\eta}T^{1/4})}{\eta} \\ &= O\left(\frac{D}{\sqrt{\eta}}T^{3/4}\right) \end{aligned} \tag{44}$$

Note the convergence rate $O(T^{3/4})$ for cumulative violations is sufficient to establish the asymptotic aggregate fairness guarantee, as $\frac{T^{3/4}}{T} = T^{-1/4} \to 0$. Methods to derive faster convergence rates are left for future work.

**Step 3: Aggregate fairness guarantee**

Recall $w_t(\hat{y}_{b_t}) = |b_t|(v_t(\hat{y}_{b_t}) - \epsilon)$. From (44):

$$\sum_{t=1}^{T} |b_t| v_t(\hat{y}_{b_t}) = \sum_{t=1}^{T} w_t(\hat{y}_{b_t}) + \epsilon \sum_{t=1}^{T} |b_t| \leq O(T^{3/4}) + \epsilon N_T, \tag{45}$$

where $N_T := \sum_{t=1}^{T} |b_t|$ is the total number of samples.

Dividing by $N_T$ and using $N_T \geq b_{\min} T$ for some $b_{\min} > 0$:

$$\begin{aligned}\frac{1}{N_T}\sum_{t=1}^{T} |b_t| v_t(\hat{y}_{b_t}) &\leq \epsilon + \frac{O(T^{3/4})}{N_T} \\ &\leq \epsilon + O\left(\frac{T^{3/4}}{b_{\min}T}\right) \\ &= \epsilon + O(T^{-1/4}) \end{aligned} \tag{46}$$

Then:

$$\limsup_{T\to\infty} \frac{1}{\sum_{t=1}^{T}|b_t|} \sum_{t=1}^{T} |b_t| v_t(\hat{y}_{b_t}) \leq \epsilon + \lim_{T\to\infty} O(T^{-1/4}) = \epsilon$$

Which completes the proof. □

### H.3. Proof of Theorem 4.1

Below we provide a proof using standard methods from convex analysis.

*Proof.* **Step 1: Existence and uniqueness of the minimizer.** Fix $z \in \mathbb{R}^n$ and define $f_z(y) := h(y) - \langle z, y\rangle$. Take subgradient $v_0 \in \partial h(0)$. By $\mu$-strong convexity,

$$h(y) \geq h(0) + \langle v_0, y\rangle + \frac{\mu}{2}\|y\|_2^2.$$

Subtracting $\langle z, y\rangle$ gives

$$f_z(y) = h(y) - \langle z, y\rangle \geq h(0) + \frac{\mu}{2}\|y\|_2^2 + \langle v_0 - z, y\rangle.$$

Using Cauchy-Schwarz, we find that

$$f_z(y) \geq h(0) + \frac{\mu}{2}\|y\|_2^2 - \|v_0 - z\|_2 \, \|y\|_2 \to +\infty \quad \text{as } \|y\|_2 \to \infty$$

Hence $f_z$ is coercive. Since $h$ is strongly convex (and therefore continuous) and $-\langle z, y\rangle$ is continuous, $f_z$ is continuous. By the Weierstrass extreme value theorem, a continuous coercive function on a closed set attains its minimum. Therefore, $f_z$ attains its minimum. Additionally, strong convexity of $h$ implies that $f_z$ is $\mu$-strongly convex. Strongly convex functions have at most one minimizer, so $g(z)$ is unique.

**Step 2: Lipschitz continuity** The minimizer $g(z)$ in the statement satisfies (Rockafellar, 2015, Thm. 27.4)

$$\begin{aligned}0 \in \partial h(g(z)) - z + N_{\mathcal{C}}(g(z)) \iff \\ z \in \partial h(g(z)) + N_{\mathcal{C}}(g(z)),\end{aligned}$$

where $N_{\mathcal{C}}(y)$ is the normal cone to $\mathcal{C}$ at $y$. Let $z, q \in \mathbb{R}^n$ with minimizers $g(z), g(q)$. Then there exist $v_z \in \partial h(g(z)), v_q \in \partial h(g(q)), n_z \in N_{\mathcal{C}}(g(z)), n_q \in N_{\mathcal{C}}(g(q))$ such that

$$z = v_z + n_z, \quad q = v_q + n_q.$$

Given these equalities for $z$ and $q$, we have:

$$\begin{aligned}&\langle z - q, g(z) - g(q)\rangle = \\ &\quad \langle v_z - v_q, g(z) - g(q)\rangle + \langle n_z - n_q, g(z) - g(q)\rangle.\end{aligned}$$

The normal cone operator is monotone, so $\langle n_z - n_q, g(z) - g(q)\rangle \geq 0$. Since $\partial h$ is $\mu$-strongly monotone,

$$\langle v_z - v_q, g(z) - g(q)\rangle \geq \mu\|g(z) - g(q)\|_2^2.$$

Thus

$$\langle z - q, g(z) - g(q)\rangle \geq \mu\|g(z) - g(q)\|_2^2.$$

By Cauchy-Schwarz and dividing by $\|g(z) - g(q)\|_2$,

$$\|g(z) - g(q)\|_2 \leq \frac{1}{\mu}\|z - q\|_2,$$

so $g$ is globally Lipschitz with constant $1/\mu$.

**Step 3: Differentiability almost everywhere.** By Rademacher's theorem, every Lipschitz map $\mathbb{R}^n \to \mathbb{R}^n$ is differentiable almost everywhere. Hence $g$ is differentiable almost everywhere. □

### H.4. Proof of Corollary 4.2

*Proof.* By Step 2 in Theorem 4.1, $g$ is $1/\mu$-Lipschitz continuous:

$$\|g(z_1) - g(z_2)\|_2 \leq \frac{1}{\mu}\|z_1 - z_2\|_2 \quad \forall z_1, z_2 \in \mathcal{C} \tag{47}$$

Note that if a function $g : \mathbb{R}^n \to \mathbb{R}^n$ is $1/\mu$-Lipschitz continuous, then at any point where $g$ is differentiable, the operator norm of the Jacobian satisfies $\|\mathbf{D}g(z)\|_{op} \leq \frac{1}{\mu}$. Since $g$ is $1/\mu$-Lipschitz and differentiable almost everywhere (by Theorem 4.1), we have $\|\mathbf{D}g(z)\|_2 \leq \frac{1}{\mu}$ almost everywhere. By the chain rule, $\nabla_X(g \circ f)(X) = \mathbf{D}g(f(X)) \cdot \nabla_X f(X)$. Therefore:

$$\begin{aligned}\|\nabla_X(g \circ f)(X)\|_2 &\leq \|\mathbf{D}g(f(X))\|_2 \cdot \|\nabla_X f(X)\|_2 \\ &\leq \frac{1}{\mu}\|\nabla_X f(X)\|_2\end{aligned}$$

□

### H.5. Proof of Theorem 4.3

*Proof.* (1): We first show that $R_\mathcal{I}$ is a polyhedron. By the KKT conditions (9), $z \in R_\mathcal{I}$ if and only if there exist $y \in \mathbb{R}^n$ and $\lambda \in \mathbb{R}^m$ satisfying:

$$\nabla \mathcal{L}_y = y - z + A^\top \lambda = 0, \tag{48}$$
$$(Ay)_i = b_i \quad \forall i \in \mathcal{I}, \tag{49}$$
$$(Ay)_i < b_i \quad \forall i \notin \mathcal{I}, \tag{50}$$
$$\lambda_i > 0 \quad \forall i \in \mathcal{I}, \tag{51}$$
$$\lambda_i = 0 \quad \forall i \notin \mathcal{I}. \tag{52}$$

From (48) and (52), we have $y = z - A_\mathcal{I}^\top \lambda_\mathcal{I}$, where $\lambda_\mathcal{I}$ contains entries indexed by $\mathcal{I}$. Then, plugging into (49) we have:

$$A_\mathcal{I} z - A_\mathcal{I} A_\mathcal{I}^\top \lambda_\mathcal{I} = b_\mathcal{I} \tag{53}$$
$$\implies \lambda_\mathcal{I} = (A_\mathcal{I} A_\mathcal{I}^\top)^+ (A_\mathcal{I} z - b_\mathcal{I}) > 0 \quad \text{(by (51))} \tag{54}$$

Note this is a system of linear inequalities in $z$ since the pseudoinverse is a linear operator.

For the inactive constraints, note first that substituting (54) into (48) gives:

$$\begin{aligned} y &= z - A_\mathcal{I}^\top (A_\mathcal{I} A_\mathcal{I}^\top)^+ (A_\mathcal{I} z - b_\mathcal{I}) \\ &= \underbrace{\left[I - A_\mathcal{I}^\top (A_\mathcal{I} A_\mathcal{I}^\top)^+ A_\mathcal{I}\right]}_{P_\mathcal{I}} z + \underbrace{A_\mathcal{I}^\top (A_\mathcal{I} A_\mathcal{I}^\top)^+ b_\mathcal{I}}_{c_\mathcal{I}}. \end{aligned} \tag{55}$$

The inactive constraint conditions $(Ay)_j < b_j$ for $j \notin \mathcal{I}$ become:

$$(AP_\mathcal{I} z + Ac_\mathcal{I})_j < b_j \quad \forall j \notin \mathcal{I}, \tag{56}$$

which are $m - |\mathcal{I}|$ linear inequalities in $z$. Therefore, $R_\mathcal{I}$ is defined by the finite system of linear inequalities (54) and (56), making it a polyhedron.

To show the regions partition $\mathbb{R}^n$, note that for any $z \in \mathbb{R}^n$, the quadratic program has a unique solution $g(z)$ by strong convexity with a unique active set $\mathcal{A}(z)$ and KKT multipliers $\lambda(z)$. If strict complementarity holds at $z$ (i.e., $\lambda_i(z) > 0$ for all $i \in \mathcal{A}(z)$), then $z \in R_{\mathcal{A}(z)}$. Points where strict complementarity fails lie on boundaries between regions, and such boundaries have measure zero (as shown below). Therefore, the regions $\{R_\mathcal{I}\}$ partition $\mathbb{R}^n$ except for a set of measure zero.

(2): Fix $z \in R_\mathcal{I}$. From the KKT stationarity condition, $g(z) = z - A_\mathcal{I}^\top \lambda_\mathcal{I}(z)$, and from the active constraints $A_\mathcal{I} g(z) = b_\mathcal{I}$, we obtain (as shown in part (1)):

$$\lambda_\mathcal{I}(z) = (A_\mathcal{I} A_\mathcal{I}^\top)^+ (A_\mathcal{I} z - b_\mathcal{I}). \tag{57}$$

Substituting:

$$\begin{aligned} g(z) &= z - A_\mathcal{I}^\top (A_\mathcal{I} A_\mathcal{I}^\top)^+ (A_\mathcal{I} z - b_\mathcal{I}) \\ &= z - A_\mathcal{I}^\top (A_\mathcal{I} A_\mathcal{I}^\top)^+ A_\mathcal{I} z + A_\mathcal{I}^\top (A_\mathcal{I} A_\mathcal{I}^\top)^+ b_\mathcal{I} \\ &= \left[I - A_\mathcal{I}^\top (A_\mathcal{I} A_\mathcal{I}^\top)^+ A_\mathcal{I}\right] z + A_\mathcal{I}^\top (A_\mathcal{I} A_\mathcal{I}^\top)^+ b_\mathcal{I}. \end{aligned} \tag{58}$$

Setting $P_\mathcal{I} = I - A_\mathcal{I}^\top (A_\mathcal{I} A_\mathcal{I}^\top)^+ A_\mathcal{I}$ and $c_\mathcal{I} = A_\mathcal{I}^\top (A_\mathcal{I} A_\mathcal{I}^\top)^+ b_\mathcal{I}$, we have

$$g(z) = P_\mathcal{I} z + c_\mathcal{I} \quad \forall z \in R_\mathcal{I}. \tag{59}$$

Since $P_\mathcal{I}$ and $c_\mathcal{I}$ depend only on $A_\mathcal{I}$ and $b_\mathcal{I}$ (which are fixed for the region $R_\mathcal{I}$), both are constant on $R_\mathcal{I}$. Therefore, $g$ is affine on $R_\mathcal{I}$.

(3): Consider two distinct regions $R_\mathcal{I}$ and $R_\mathcal{J}$ with $\mathcal{I} \neq \mathcal{J}$. The boundary between these regions consists of points $z$ where at least one of the defining inequalities of $R_\mathcal{I}$ or $R_\mathcal{J}$ becomes an equality. This occurs when:

1. A multiplier becomes zero: $\lambda_i(z) = 0$ for some $i \in \mathcal{I} \cap \mathcal{J}$, or
2. A constraint becomes active or inactive: $(Ag(z))_j = b_j$ for some $j$ where the active status changes between $\mathcal{I}$ and $\mathcal{J}$.

For case (a): By (54), $\lambda_\mathcal{I}(z)$ is an affine function of $z$ within the closure of $R_\mathcal{I}$ (since the Moore-Penrose pseudoinverse is a linear operator). The set $\{\lambda_i(z) = 0\}$ for a specific $i \in \mathcal{I}$ is therefore a hyperplane in $\mathbb{R}^n$, which has Lebesgue measure zero.

For case (b): Consider an inactive constraint index $j \notin \mathcal{I}$. Within region $R_\mathcal{I}$, we have $g(z) = P_\mathcal{I} z + c_\mathcal{I}$, so:

$$(Ag(z))_j = A_j P_\mathcal{I} z + A_j c_\mathcal{I}, \tag{60}$$

which is an affine function of $z$. The level set $\{z \in R_\mathcal{I} : (Ag(z))_j = b_j\}$ is therefore the intersection of a hyperplane with the polyhedral region $R_\mathcal{I}$, which is at most $(n-1)$-dimensional and hence has measure zero.

The complete level set $\{z \in \mathbb{R}^n : (Ag(z))_j = b_j\}$ is the union over all regions:

$$\{z : (Ag(z))_j = b_j\} = \bigcup_{\mathcal{I}} (\{z : A_j P_{\mathcal{I}} z + A_j c_{\mathcal{I}} = b_j\} \cap R_{\mathcal{I}}) \tag{61}$$

Since this is a finite union of sets, each of which is at most $(n-1)$-dimensional, the entire level set is at most $(n-1)$-dimensional and has Lebesgue measure zero in $\mathbb{R}^n$. □

### H.6. Proof of Theorem 4.4

*Proof.* We first derive the explicit form of the Jacobian before proving it is an orthogonal projection matrix. Lastly, we establish the stated spectral properties.

The optimization problem defining $g(z)$ can be written as

$$g(z) = \arg\min_{y \in \mathcal{C}} \left\{ \frac{1}{2}\|y\|^2 - \langle z, y \rangle \right\}. \tag{62}$$

The Lagrangian associated with (8) is as follows:

$$\mathcal{L}(y, \lambda) = \frac{1}{2}\|y\|_2^2 - z^\top y + \lambda^\top (Ay - b), \tag{63}$$

where $\lambda \in \mathbb{R}^m$ are the Lagrange multipliers. Since $h(y) = \frac{1}{2}\|y\|_2^2$ is 1-strongly convex and $\mathcal{C}$ is a nonempty closed convex set, by Step 1 in Theorem 4.1, the minimizer $g(z)$ is unique. Additionally, the KKT conditions are necessary and sufficient for optimality. Therefore, there exists a unique pair $(g(z), \lambda(z)) \in \mathbb{R}^n \times \mathbb{R}^m$ satisfying the KKT conditions.

By the complementarity slackness KKT condition, we have $\lambda_i(z) > 0$ if and only if the constraint is active, i.e., $(Ag(z))_i = b_i$ so $i \in \mathcal{A}(z)$. Therefore, $\lambda_i(z) = 0$ for all $i \notin \mathcal{A}(z)$.

From the stationarity KKT condition, we have

$$g(z) = z - A^\top \lambda(z). \tag{64}$$

Since $\lambda_i(z) = 0$ for $i \notin \mathcal{A}(z)$, we can write $A^\top \lambda(z) = A_{\mathcal{A}}^\top \lambda_{\mathcal{A}}(z)$, where $\lambda_{\mathcal{A}}(z) \in \mathbb{R}^{|\mathcal{A}|}$ only contains Lagrange multipliers indexed by the active constraint set.

For the active constraints, we have $(Ag(z))_i = b_i$ for all $i \in \mathcal{A}(z)$. In matrix form:

$$A_{\mathcal{A}} g(z) = b_{\mathcal{A}}, \tag{65}$$

where $b_{\mathcal{A}} \in \mathbb{R}^{|\mathcal{A}|}$ contains the components of $b$ indexed by $\mathcal{A}(z)$.

Substituting (64) into (65):

$$A_{\mathcal{A}} z - A_{\mathcal{A}} A_{\mathcal{A}}^\top \lambda_{\mathcal{A}}(z) = b_{\mathcal{A}}. \tag{66}$$

Using the Moore-Penrose pseudoinverse $(A_{\mathcal{A}} A_{\mathcal{A}}^\top)^+$, we obtain the least-norm solution:

$$\lambda_{\mathcal{A}}(z) = (A_{\mathcal{A}} A_{\mathcal{A}}^\top)^+ (A_{\mathcal{A}} z - b_{\mathcal{A}}). \tag{67}$$

Substituting (67) into (64):

$$\begin{aligned} g(z) &= z - A_{\mathcal{A}}^\top (A_{\mathcal{A}} A_{\mathcal{A}}^\top)^+ (A_{\mathcal{A}} z - b_{\mathcal{A}}) \\ &= z - A_{\mathcal{A}}^\top (A_{\mathcal{A}} A_{\mathcal{A}}^\top)^+ A_{\mathcal{A}} z + A_{\mathcal{A}}^\top (A_{\mathcal{A}} A_{\mathcal{A}}^\top)^+ b_{\mathcal{A}} \\ &= \left[ I - A_{\mathcal{A}}^\top (A_{\mathcal{A}} A_{\mathcal{A}}^\top)^+ A_{\mathcal{A}} \right] z + A_{\mathcal{A}}^\top (A_{\mathcal{A}} A_{\mathcal{A}}^\top)^+ b_{\mathcal{A}}. \end{aligned} \tag{68}$$

From (68), we observe that the formula for $g(z)$ depends on the active set $\mathcal{A}(z)$. By Theorem 4.3(1)-(2), the domain $\mathbb{R}^n$ is partitioned into polyhedral regions where the active set is constant, and within each such region $g$ is affine. Therefore, at points in the interior of these regions, $g$ is differentiable with Jacobian

$$\mathbf{D}g(z) = I - A_{\mathcal{A}}^\top (A_{\mathcal{A}} A_{\mathcal{A}}^\top)^+ A_{\mathcal{A}}. \tag{69}$$

Note this is the standard formula for the orthogonal projection onto $\ker(A_{\mathcal{A}})$ using the Moore-Penrose pseudoinverse. Since $P = \mathbf{D}g(z)$ is a symmetric projection matrix, all eigenvalues belong to $\{0, 1\}$: for any eigenvalue $\lambda$ with eigenvector $v \neq 0$, we have $\lambda^2 v = P^2 v = Pv = \lambda v$, so $\lambda(\lambda - 1) = 0$.

Additionally, since $Pv \in \ker(A_{\mathcal{A}})$ for all $v$, we have $A_{\mathcal{A}}(Pv) = 0$, which means $\mathbf{D}g(z) \cdot v$ lies in the tangent space to the constraint surface at $g(z)$. Geometrically, the tangent space at $g(z)$ is precisely $\ker(A_{\mathcal{A}})$, since a direction $d$ is tangent to $\mathcal{C}$ at $g(z)$ if and only if $\langle a_i, d \rangle = 0$ for all active constraint normals $a_i^\top$ (the rows of $A_{\mathcal{A}}$). Therefore, the Jacobian completely suppresses the component of any vector that is normal to the active constraint surface. □