# OpenReview forum: "Differentiable Optimization Layers for Guaranteed Fairness in Deep Learning"
_ICML.cc/2026/Conference — ICML 2026 regular_

### Official Review · Reviewer_22dP · 2026-03-09

**Soundness:** 2
**Presentation:** 3
**Significance:** 3
**Originality:** 3
**Overall Recommendation:** 4
**Confidence:** 3

**Summary:**

This paper proposes a differentiable fairness layer that is appended to a neural network’s output and projects predictions onto a fairness-constrained set during training and inference. Beyond the batch-based layer formulation, the paper also introduces an online primal-dual inference algorithm for small-batch or streaming settings, where enforcing fairness per batch can be overly restrictive. The submission combines theoretical analysis of differentiability and stability with experiments on loan default prediction, employee wage modeling, and synthetic regression, arguing that the proposed layer can provide fairness guarantees while remaining end-to-end trainable and practically effective.

**Compliance With Llm Reviewing Policy:**

Affirmed.

**Final Justification:**

I maintain my initial score.

**Key Questions For Authors:**

1. How should practitioners interpret the fairness guarantees when the paper replaces classical hard classification notions with expectation-based convex relaxations?
2. Can the authors better clarify what is genuinely new relative to prior work on differentiable optimization layers, fairness projection, and primal-dual fairness enforcement?
3. How sensitive is the online primal-dual inference procedure to the choice of step size, threshold batch size, and violation statistic in practice?

**Limitations:**

Yes.

**Strengths And Weaknesses:**

**Strengths**
1. The paper addresses an important problem at the intersection of fairness and deep learning.
2. The proposed fairness layer is conceptually clean and reasonably general.
3. The paper includes both theory and experiments, and the overall empirical story is positive.

**Weaknesses**
1. The novelty claim is somewhat overstated relative to prior differentiable optimization and fairness-constrained learning literature. The paper positions the fairness layer as the first method satisfying verified fairness, end-to-end learning, and flexibility, but this claim feels stronger than what is fully justified. The work is a meaningful combination of known ingredients—projection-based fairness enforcement, differentiable optimization layers, and primal-dual style constraint handling—but the paper could do a better job of clarifying exactly what is fundamentally new beyond adapting these tools to this setting.
2. Some key theoretical and practical assumptions are restrictive or under-discussed. A number of guarantees rely on convexity, affine constraints, strong convexity of the discrepancy function, and fairness notions expressed through expectations rather than more standard hard classification notions. The paper does acknowledge that classical notions like demographic parity and equalized odds become nonconvex in the hard-assignment form, but this also means the guarantees apply to a softened formulation rather than the original definitions many practitioners may expect.

---

> ### Author Rebuttal · Authors · 2026-03-30
>
> We thank the reviewer for their thoughtful review and for stating that the fairness layer method is both conceptually clean and reasonably general. We address each of the stated weaknesses and questions below:
>
> **1 (Novelty Claim, stated weakness 1 and question 2)**: The reviewer is correct that the individual components are established. Our contribution is the specific architectural integration and the theoretic/algorithmic mechanisms required to make the integration useful in practice. Specifically:
>
> The fairness layer as an architectural component: Prior works utilize differentiable optimization layers in predict-then-optimize pipelines where the network estimates optimization parameters. We instead apply the differentiable optimization problem directly to network outputs, which is a fundamentally different computational role that requires different, specifically tailored analysis (ex: gradient stability results in Theorem 2).
>
> The online primal-dual inference algorithm: Our algorithm provides aggregate fairness guarantees across arbitrarily small batches, which is a challenge that does not arise in the standard predict-then-optimize or post-processing settings. This requires the careful analysis of when batch-level constraints for fairness with protected attributes imply aggregate fairness.
>
> The simultaneous satisfaction of (G1)–(G3): We acknowledge that framing as "the first" method may overstate the case if interpreted too broadly. We have revised to say that, to our knowledge, no prior work provides an end-to-end trainable architectural component with verified affine constraint satisfaction that is architecture-agnostic. We have also included the following comparison table in an appendix in our updated manuscript:
>
> | **Method** | **Type** | **(G1)** | **(G2)** | **(G3)** |
> |---|---|---|---|---|
> | **Fairness Layer (ours)** | In | ✓ | ✓ | ✓ |
> | Post-hoc adjustments/projection (Wei et al., 2020; Hardt et al., 2016) | Post | ✓ | ✗ | ✓ |
> | Data reweighting / augmentation (Kamiran & Calders, 2012; Ktena et al., 2024) | Pre | ✗ | ✓ | ✓ |
> | Lagrangian penalty (Manisha & Gujar, 2020; Cotter et al., 2019) | In | ✗ | ✓ | ✓ |
> | Convex surrogates / DCP (Zafar et al., 2017; Wu et al., 2018) | In | ○ᵃ | ✓ | ✗ᵇ |
>
> ᵃ Enforces surrogate constraints of hard-thresholded problems, but can be adapted to exactly satisfy affine constraints
> ᵇ Restricted to models with convex decision boundaries
>
> **2 (Assumptions, stated weakness 2 and question 1):**  We note that no such thresholds exist in regression settings, and the exact constraints required by various laws (ex: laws mentioned in response to reviewer 1 / LugD) fit directly into the framework. Next, we note that the expectation-based formulations are not ad-hoc relaxations but rather are established fairness criteria in their own right (see, for example, Mean Difference in Calders et al. 2013, bounded group loss in Agarwal et al., 2019, among others).
>
> Practitioners should interpret our guarantees as follows: the fairness layer ensures that the model outputs satisfy affine constraints at a pre-specified level. For settings where hard classification-based constraints are required, our expectation-based constraints are a necessary but not sufficient condition. In our updated manuscript, we have added a paragraph clarifying when expectation-based constraints align with established desideratum (regression, scoring, continuous predictions) versus when they serve as a tractable proxy for various classical, classification-based notions.
>
> **3 (Primal-dual inference algorithm hyperparameters, question 3):**
> Step size schedule: The adaptive schedule follows standard online convex optimization theory, leaving only the initial scale $\eta_0$ as a free parameter. We found in practice that the performance is robust across a range of $\eta_0$ values; we used $\eta_0 = 0.1$ for the new, large-scale vision experiments and $\eta_0 = 0.5$ for the synthetic experiments without tuning per-dataset. By the convergence rate in Theorem 3, the asymptotic guarantee is independent of $\eta_0$; the initial scale affects only the behavior in the first several batches.
>
> Threshold batch size $b_\tau$: This parameter choice is driven by practical criterion: $b_\tau$ should be large enough that each batch reliably contains at least one observation per protected group (so that per-batch group statistics are well-defined). The synthetic experiments explicitly test three batch-size regimes, demonstrating consistent F-Layer performance across the transition between regimes. In practice, setting $b_\tau$ to a moderate multiple of the number of protected groups is a reasonable default.
>
> Violation statistic: We use the batch size-weighted gap $w_t = |b_t| \cdot (v_t - \varepsilon)$, which is the natural choice given our constraints. The $|b_t|$ weighting ensures that larger batches contribute proportionally to the aggregate violation, which is the quantity bounded in Theorem 3.

---

> > ### Author Rebuttal · Reviewer_22dP · 2026-04-03
> >
> > Thank you for the rebuttal, and I will maintain my original positive score.

---

### Official Review · Reviewer_k6Mz · 2026-03-11

**Soundness:** 3
**Presentation:** 2
**Significance:** 2
**Originality:** 2
**Overall Recommendation:** 4
**Confidence:** 2

**Summary:**

This paper proposes a differentiable “fairness layer” that can be appended to the output of a neural network to enforce group fairness constraints directly during prediction by projecting model outputs onto a fairness-constrained optimization set.

**Compliance With Llm Reviewing Policy:**

Affirmed.

**Final Justification:**

n/a

**Key Questions For Authors:**

1 - recheck the Eq.8.

2 - Theorems 4.3 and 4.4 could have been merged.

3 - The paper is well-developed and offers some interesting insights. However, it doesn't demonstrate the significance thoroughly, as it lacks sufficient numerical ablation studies on large, realistic datasets and complex architectures. Additionally, I would like to see how adding a layer to the architecture impacts the computational cost and the training process overall and during parameter tuning. The effects may vary depending on the type of architecture, which the authors did not discuss in detail. The paper overall does not include a discussion on architecture.

**Strengths And Weaknesses:**

Soundness: The paper appears technically sound.

Presentation: The presentation is lacking and has several flaws. For instance, they define 'd' for different concepts (dimension and function), and they do the same for 'h'. Some theorems could be consolidated, and certain proofs should be incorporated into the main text for better clarity.

Significance: The work addresses an important problem, but the authors did not provide sufficient empirical validation.

Originality: The paper introduces a novel formulation and an associated optimization method.

---

> ### Author Rebuttal · Authors · 2026-03-30
>
> We thank the reviewer for their thoughtful review and for acknowledging the novelty of our method. We address each of the stated weaknesses and questions below:
>
> **1 (Presentation Fixes):** We have updated our manuscript to ensure $d$ is not used to define both the input dimension and the discrepancy function, and that $h$ is not used in multiple ways in different parts of the manuscript.
>
> We agree that the setup for Theorems 4.3 and 4.4 are identical and we have therefore combined them. While we unfortunately cannot include entire proofs in the main paper (as this would exceed the 8-page limit), we have added summary paragraphs below theorems describing proof techniques used to increase clarity.
>
> Lastly, the equality sign in Eq. 8 been replaced with an equivalence arrow to avoid confusion:
>
> $\sum_{t=1}^T w_t(\hat y_{b_t}) \le 0
> \qquad \Longleftrightarrow \qquad
> \frac{1}{\sum_{t=1}^T |b_t|}
> \sum_{t=1}^T |b_t| v(t) \le \epsilon$
>
> **2 (Experiment Limitations):** We are excited to share results from additional experiments on larger datasets with complex architectures. We evaluate two tasks: (1) CelebA dataset (200k+ images): predicting whether a person is smiling, constrained so average predicted logits are similar across race/gender groups. This is motivated by smile detection in smartphone cameras, where disparate accuracy across demographics means some groups consistently get worse auto-captured photos. (2) FairFace dataset (100k+ images): predicting age (over/under 30) with constraints requiring similar average predictions across intersectional race/gender groups. This is motivated by age-gated access systems (e.g., content filtering, retail verification), where systematic prediction differences could deny access to certain demographics. For each task/fairness method, we train across five architectures via GPU: vision transformer ViT-B/16 with LoRA adapters (86M parameters), Swin-T (shifted-window hierarchical transformer), DenseNet-121 (dense convolutional network), ResNet-18 (residual network), and a convolutional network trained from scratch. The tables below display results (all methods satisfy constraints in both tasks). The F-Layer consistently improves accuracy over all baselines. Relative gains over the Projection method are typically $2–5$%, while gains over Lagrangian methods are much larger due to training sensitivity despite cross-validated hyperparameter selection.
>
> **CelebA test set results across architectures,** ΔAcc shows percent change in **relative** (not absolute) accuracy to the F-Layer:
>
> | Backbone | Method | Accuracy | ΔAcc (%) |
> |:---------|:-------|:--------:|:--------:|
> | ViT-B/16 (LoRA) | F-Layer | 0.9062 | — |
> |  | Projection | 0.8760 | -3.45 |
> |  | Penalty | 0.5014 | -80.74 |
> |  | Strict Penalty | 0.4944 | -83.31 |
> | DenseNet-121 | F-Layer | 0.9149 | — |
> |  | Projection | 0.8930 | -2.46 |
> |  | Penalty | 0.5014 | -82.47 |
> |  | Strict Penalty | 0.4916 | -86.09 |
> | Swin-T | F-Layer | 0.9177 | — |
> |  | Projection | 0.8882 | -3.32 |
> |  | Penalty | 0.5014 | -83.02 |
> |  | Strict Penalty | 0.4964 | -84.89 |
> | ResNet-18 | F-Layer | 0.9038 | — |
> |  | Projection | 0.8852 | -2.10 |
> |  | Penalty | 0.4997 | -80.85 |
> |  | Strict Penalty | 0.4922 | -83.62 |
> | CustomCNN  | F-Layer | 0.8141 | — |
> |  | Projection | 0.8088 | -0.65 |
> |  | Penalty | 0.5014 | -62.35 |
> |  | Strict Penalty | 0.4920 | -65.48 |
>
>
> **FairFace test set results across architectures:** ΔAcc shows percent change in **relative** (not absolute) accuracy to the F-Layer:
>
> | Backbone | Method | Accuracy | ΔAcc (%) |
> |:---------|:-------|:--------:|:--------:|
> | ViT-B/16 (LoRA) | F-Layer | 0.7749 | — |
> |  | Projection | 0.7541 | -2.76 |
> |  | Penalty | 0.5496 | -41.01 |
> |  | Strict Penalty | 0.5341 | -45.08 |
> | DenseNet-121 | F-Layer | 0.7623 | — |
> |  | Projection | 0.7389 | -3.17 |
> |  | Penalty | 0.5506 | -38.44 |
> |  | Strict Penalty | 0.5352 | -42.43 |
> | Swin-T | F-Layer | 0.8081 | — |
> |  | Projection | 0.7646 | -5.69 |
> |  | Penalty | 0.5506 | -46.75 |
> |  | Strict Penalty | 0.5359 | -50.79 |
> | ResNet-18 | F-Layer | 0.7616 | — |
> |  | Projection | 0.7479 | -1.84 |
> |  | Penalty | 0.5496 | -38.59 |
> |  | Strict Penalty | 0.5306 | -43.55 |
> | CustomCNN | F-Layer | 0.6729 | — |
> |  | Projection | 0.6703 | -0.39 |
> |  | Penalty | 0.5496 | -22.44 |
> |  | Strict Penalty | 0.5346 | -25.86 |
>
> In terms of computation times: for the CelebA datasets (where 6 constraints are present), the F-Layer method resulted in an average of around $13$% higher computation time (in s) per epoch on average. This overhead is therefore not prohibitive in using the method. As the number of constraints grows, however, we expect larger computational overhead relative to standard training as the convex QP grows in complexity.
>
> In our updated manuscript, we have included a detailed description of these results and a discussion on the architecture-agnostic nature of our method; it can be appended to the output layer of any network to project onto an affine constraint set.

---

> > ### Author Rebuttal · Reviewer_k6Mz · 2026-04-04
> >
> > I will increase my score. Thank you

---

### Official Review · Reviewer_LugD · 2026-03-11

**Soundness:** 3
**Presentation:** 3
**Significance:** 2
**Originality:** 2
**Overall Recommendation:** 4
**Confidence:** 3

**Summary:**

The paper proposes a method to enforce fairness in deep learning models by introducing a differentiable optimization layer at the output of a neural network. This layer solves a constrained optimization problem that adjusts the model’s predictions so that predefined fairness constraints are satisfied while maintaining predictive accuracy. Because the optimization layer is differentiable, the entire model can be trained end-to-end using backpropagation. The authors also propose an online primal–dual algorithm to ensure fairness over a sequence of predictions rather than within individual batches. Experimental results show that the method effectively enforces fairness constraints while maintaining competitive model performance.

**Compliance With Llm Reviewing Policy:**

Affirmed.

**Final Justification:**

I am satisfied with the authors' response.

**Key Questions For Authors:**

Please see my comments on the weakness of this work.

**Limitations:**

Please see my comments on the weakness of this work.

**Strengths And Weaknesses:**

Strength:
1- The concept of a fairness layer is new. Instead of regularization-based fairness, the paper proposed to utilize a fairness layer.
2- As far as I know, many fairness methods only encourage fairness. The proposed method in this paper guarantees fairness constraint satisfaction.
3- The paper has rigorous theoretical guarantees to show differentiability, constraint satisfaction. I believe this makes this work proper for ICML.

Weakness:
1- The work only considered group fairness metrics. It is not straightforward to me if authors can argue that by utilizing the fairness layer, we can guarantee other types of fairness criterion.
2- The numerical simulations are limited to small or medium datasets. The evaluations did not provide: 1- large-scale datasets, and 2- complex deep architectures.
3- The authors did not clarify how they ended up in utilizing the set of fairness constraints in the numerical results section. I believe that such constraints and the threshold provided requires an expert. This makes the proposed approach less appealing in practice.

---

> ### Author Rebuttal · Authors · 2026-03-30
>
> We thank the reviewer for their thoughtful review and for stating that the rigorous guarantees of our method make the work suitable for ICML. We address each of the stated weaknesses below:
>
> **1 (Other Notions of Fairness):** The reviewer is correct that we focus on notions of group fairness in this work; when we state that the Fairness Layer extends to other notions of fairness, we mean other notions of group fairness that were not explicitly listed in the current paper, including bounded group loss (Agarwal et al., 2019), for example. We have clarified this point in our updated manuscript.
>
> Note, however, that the Fairness Layer formulation as stated does already encompass individual fairness, which is the other primary notion of fairness in the literature. Individual fairness reduces to imposing a Lipschitz-type constraint: given a constant $L$ and distance metric $d_\mathcal{X}(\cdot, \cdot)$, individual fairness requires that for any two predictions $\hat{y_i}$ and $\hat{y_j}$ with corresponding inputs $x_i$ and $x_j$ :
>
> $|\hat{y_i} - \hat{y_j}| \leq L \cdot d_\mathcal{X}(x_i,x_j) \; \ \forall (i,j) $
>
> By decomposing the absolute value into two separate inequalities, we are left with linear constraints in $\hat{y}$, which when combined with minimizing our standard discrepancy function fits exactly into our framework without modifications to backpropagation methods, forward pass solvers, etc.
>
> We choose to focus on group fairness constraints in this work for two reasons: first, prior works have already developed methods to impose exact individual fairness constraints in deep learning by constraining the global Lipschitz constant of the network via spectral normalization, which provides a sufficient condition for individual fairness under $\ell_p$ input metrics. Conversely, no such methods have been developed for group fairness. Second, we focus on group fairness metrics since they naturally enable/require the fairness metric to be tracked across batches. Therefore, one can study when satisfying batch-level constraints implies population-level constraints are satisfied; this lead to the natural formulation of the primal-dual inference algorithm and small/large inference batch regimes. Individual fairness constraints can be enforced within a given batch via the Fairness Layer, but careful treatment is required to ensure equation aggregate fairness holds across batches if that is the modeler's goal. We plan to explicitly develop methods for this in future works, and we have made this point clearer in our updated manuscript.
>
> **2 (Experiment Limitations):**  Thank you for pointing out this limitation. We have conducted an additional large suite of GPU-enabled experiments which compares all of the described fairness methods across $5$ different model architectures (including multiple vision transformers, convolutional neural networks, and large residual-networks) for two new datasets/tasks, one of which contains over $100,000$ images and another which contains over $200,000$ images. Please see our response to reviewer 2 (i.e. reviewer k6Mz) for a detailed discussion.
>
> **3 (Picking Constraint/Thresholds):** We agree that domain experts are often consulted/required to formulate what constraints and thresholds are most applicable for a particular setting. However, fairness constraints and associated thresholds are also instead often prescribed by law rather than a domain expert. For example, the recent EU Pay Transparency Directive (Directive 2023/970) dictates that companies must not have differences in salary exceeding $5$% for gender within the same job category. In this case, the mean demographic parity formulation with $\epsilon = .05$ is explicit. Similarly, EU Directive 2004/113/EC requires that insurance premiums must be equal across gender. This demographic parity formulation with $\epsilon =0$ is also encompassed by our formulation.
>
> Our numerical experiments aimed to demonstrate reasonable scenarios that one could incorporate fairness constraints for as opposed to a deep-dive regarding what different domain experts recommend. For example, no gender attribute exists for companies in the loan recommendation experiment setting, but one could reasonably consider a scenario in which a new law dictates loan companies must not discriminate against rural vs. non-rural business or new vs. old businesses. For our experiments, we chose thresholds small enough such that the unconstrained optimal solution does not already lie in the feasible set. We also vary the thresholds in our synthetic experiment ablation study to reflect the utility of the Fairness Layer method across the tightness of thresholds.

---

> > ### Author Rebuttal · Reviewer_LugD · 2026-04-03
> >
> > I am satisfied with the authors' response to my concerns. Is it possible that the authors provide an ablation study on the threshold parameters for other cases except the synthetic case? I will increase my score accordingly

---

> > > ### Author Response · Authors · 2026-04-07
> > >
> > > We thank the reviewer for their willingness to consider raising their score, and we are excited by the results of our additional ablation study.
> > >
> > > We extend the ablation study of the two thresholds in our synthetic experiments to our image experiments (see rebuttal to reviewer k6Mz for dataset/task details), varying both the constraint tolerance threshold $\epsilon$ and the primal-dual threshold $b_{\tau}$.
> > >
> > > ## Ablation 1: Constraint Tightness Threshold ($\epsilon$)
> > > To ensure our original results are not simply a byproduct of a specific constraint tightness, we relaxed the allowed slack from $10^{-3}$ to $10^{-2}$ for the CelebA experiments and from $10^{-4}$ to $10^{-3}$ for the FairFace data experiments. Overall we see the same pattern of results as the original experiments: the Fairness Layer consistently results in accuracy improvements over Projection method from around $1$% to $5$% regardless of architecture and dataset. If the unconstrained and constrained solutions are close in distance (i.e. constraint slack is extremely loose), one would expect F-Layer and Projection methods to achieve more similar results as neither projection would alter the unconstrained solution.
> > >
> > > ### CelebA Experiments Results for larger slack. ΔAcc (%) shows percent change in relative (not absolute) accuracy to the F-Layer:
> > >
> > > | Backbone | Method | Accuracy | ΔAcc (%) |
> > > |:---------|:-------|:--------:|:--------:|
> > > | ViT-B/16 (LoRA) | F-Layer | 0.9067 | — |
> > > |  | Projection | 0.8606 | -5.08 |
> > > |  | Penalty | 0.5224 | -73.57 |
> > > | DenseNet-121 | F-Layer | 0.8978 | — |
> > > |  | Projection | 0.8743 | -2.62 |
> > > |  | Penalty | 0.5014 | -79.07 |
> > > | Swin-T | F-Layer | 0.8952 | — |
> > > |  | Projection | 0.8709 | -2.71 |
> > > |  | Penalty | 0.6450 | -38.80 |
> > > | ResNet-18 | F-Layer | 0.9041 | — |
> > > |  | Projection | 0.8695 | -3.83 |
> > > |  | Penalty | 0.7048 | -28.27 |
> > > | CustomCNN | F-Layer | 0.8147 | — |
> > > |  | Projection | 0.7942 | -2.52 |
> > > |  | Penalty | 0.5014 | -62.47 |
> > >
> > > ### FairFace Experiments Results for larger slack. ΔAcc (%) shows percent change in relative (not absolute) accuracy to the F-Layer:
> > >
> > > | Backbone | Method | Accuracy | ΔAcc (%) |
> > > |:---------|:-------|:--------:|:--------:|
> > > | ViT-B/16 (LoRA) | F-Layer | 0.7772 | — |
> > > |  | Projection | 0.7479 | -3.77 |
> > > |  | Penalty | 0.5506 | -29.15 |
> > > | DenseNet-121 | F-Layer | 0.7394 | — |
> > > |  | Projection | 0.7389 | -0.068 |
> > > |  | Penalty | 0.5506 | -25.53 |
> > > | Swin-T | F-Layer | 0.7735 | — |
> > > |  | Projection | 0.7646 | -1.15 |
> > > |  | Penalty | 0.5506 | -28.81 |
> > > | ResNet-18 | F-Layer | 0.7653 | — |
> > > |  | Projection | 0.7431 | -2.90 |
> > > |  | Penalty | 0.5506 | -28.04 |
> > > | CustomCNN | F-Layer | 0.6697 | — |
> > > |  | Projection | 0.6629 | -1.02 |
> > > |  | Penalty | 0.5506 | -17.77 |
> > >
> > > ## Ablation 2: Primal-Dual Threshold ($b_{\tau}$)
> > > Note that when inference batch size $b_{infer}$ is less than or equal to threshold $b_\tau$, the primal-dual inference algorithm is used; otherwise, batch-level projections are performed. The tables below show that the choice of $b_{\tau}$ has negligible impact on accuracy for moderate batch sizes (i.e. for any model, row $b_\tau = 4$ and row $b_\tau = 1024$ are similar), and that at very small batch sizes (comparable to the number of protected groups), the primal-dual algorithm, i.e. rows with $b_\tau = 4$, can yield modest accuracy gains over hard projection, consistent with its softer enforcement allowing individual batches more expressivity. When stratified sampling is used -- as is used in this ablation study --, both regimes achieve aggregate fairness (Lemma 3.1); in streaming settings where stratified sampling is infeasible, Theorem 3.2 guarantees that the sample-weighted average per-batch fairness violation still converges to at most $\epsilon$.
> > >
> > > | Backbone | $b_{\tau}$ | $b_{infer}$=14  (14 = Num. Groups) |$b_{infer}$=16 | $b_{infer}$=32 | $b_{infer}$=64 | $b_{infer}$=128 | $b_{infer}$=256 |
> > > |:---------|:----------:|:--------:|:--------:|:--------:|:--------:|:--------:|:--------:|
> > > | ResNet-18 | 4 | 0.594 | 0.615 | 0.704 | 0.748 | 0.765 | 0.769 |
> > > |  | 1024 | 0.576 | 0.607 | 0.700 | 0.748 | 0.765 | 0.769 |
> > > | ViT-B/16 (LoRA) | 4 | 0.587 | 0.606 | 0.717 | 0.758 | 0.777 | 0.784 |
> > > |  | 1024 | 0.579 | 0.607 | 0.717 | 0.758 | 0.777 | 0.784 |
> > > | CustomCNN | 4 | 0.571 | 0.577 | 0.648 | 0.668 | 0.670 | 0.680 |
> > > |  | 1024 | 0.569 | 0.577 | 0.648 | 0.668 | 0.670 | 0.680 |
> > >
> > >
> > > ### CelebA: Accuracy across inference regimes
> > >
> > > | Backbone | $b_{\tau}$ | $b_{infer}$=4 | $b_{infer}$=8 | $b_{infer}$=16 | $b_{infer}$=32 | $b_{infer}$=64 | $b_{infer}$=128 |
> > > |:---------|:----------:|:--------:|:--------:|:--------:|:--------:|:--------:|:--------:|
> > > | ResNet-18 | 4 | 0.765 | 0.778 | 0.827 | 0.862 | 0.877 | 0.889 |
> > > |  | 1024 | 0.751 | 0.775 | 0.825 | 0.862 | 0.877 | 0.889 |
> > > | ViT-B/16 (LoRA) | 4 | 0.767 | 0.783 | 0.832 | 0.865 | 0.880 | 0.894 |
> > > |  | 1024 | 0.752 | 0.781 | 0.831 | 0.865 | 0.880 | 0.893 |
> > > | CustomCNN | 4 | 0.705 | 0.714 | 0.754 | 0.781 | 0.789 | 0.801 |
> > > |  | 1024 | 0.702 | 0.714 | 0.753 | 0.781 | 0.788 | 0.801 |

---

### Decision · Program_Chairs · 2026-04-30

**Decision:**

Accept (regular)

**Comment:**

The paper proposes a differentiable layer to enforce the group fairness. While the reviewers had concerns about the limited applicability (only to group fairness metrics), clarity, and limited experiments, the authors have adequately addressed them during the discussion phase. Thus, AC suggests accepting this paper. Please make sure all additional discussions and experiments are incorporated in the final version.